# Fetal whole heart blood flow imaging using 4D cine MRI

Thomas A. Roberts [1,4 ✉], Joshua F. P. van Amerom [1,4], Alena Uus[1], David F. A. Lloyd [1,2], Milou P. M. van Poppel [1,2], Anthony N. Price[1], Jacques-Donald Tournier[1], Chloe A. Mohanadass [1], Laurence H. Jackson[1], Shaihan J. Malik[1], Kuberan Pushparajah[1,2], Mary A. Rutherford [1,3], Reza Rezavi[1,2], Maria Deprez [1] & Joseph V. Hajnal[1]

Prenatal detection of congenital heart disease facilitates the opportunity for potentially life-saving care immediately after the baby is born. Echocardiography is routinely used for screening of morphological malformations, but functional measurements of blood flow are scarcely used in fetal echocardiography due to technical assumptions and issues of reliability. Magnetic resonance imaging (MRI) is readily used for quantification of abnormal blood flow in adult hearts, however, existing in utero approaches are compromised by spontaneous fetal motion. Here, we present and validate a novel method of MRI velocity-encoding combined with a motion-robust reconstruction framework for four-dimensional visualization and quantification of blood flow in the human fetal heart and major vessels. We demonstrate simultaneous 4D visualization of the anatomy and circulation, which we use to quantify flow rates through various major vessels. The framework introduced here could enable new clinical opportunities for assessment of the fetal cardiovascular system in both health and disease.

[1] School of Biomedical Engineering & Imaging Sciences, King's College London, London SE1 7EH, UK. [2] Department of Congenital Heart Disease, Evelina Children's Hospital, London SE1 7EH, UK. [3] Centre for the Developing Brain, King's College London, London SE1 7EH, UK. [4] These authors contributed equally: Thomas A. Roberts and Joshua F. P. van Amerom. ✉email: t.roberts@kcl.ac.uk

**B**lood flow imaging of the fetal heart and great vessels is extremely challenging because the heart beats rapidly, the vessels are small and the fetus is prone to spontaneous movement as well as displacements caused by maternal respiration. Pulsed wave Doppler echocardiography[1–3] is readily available in clinics for measuring blood flow rates, and is very low cost compared to using magnetic resonance imaging (MRI), but in practice is not routinely used in fetal or pediatric cardiology. Many reports have documented the errors associated with ultrasound-based volumetric flow estimates[4–8]: Doppler echocardiography relies on line-of-sight velocity assessment combined with assumptions about the shape and flow profile of the targeted blood vessel to convert measurements to flow rates, and can only provide Doppler velocities at a single point within the vessel lumen. Furthermore, the accuracy, precision, and reliability of Doppler measurements also depend on fetal lie, maternal habitus, and the level of expertize of the operating sonographer.

Phase contrast (PC) MRI methods are routinely used for time-resolved quantitative blood flow measurements of the cardiovascular system in adult hearts[9–12], and to a lesser extent in pediatric populations[13–15] and in neonates[16,17]. Two-dimensional (2D) PC-MRI typically measures through-plane blood flow whereas four-dimensional (4D flow) PC-MRI[9–12] is a volumetric method for capturing a temporally dynamic three-directional velocity vector field of blood flow. 4D flow MRI is widely performed in adults where the imaging volume may encompass the whole heart or it may be prescribed to study flow in a specific region or vessel.

In the last decade, some developments have been made toward using PC-MRI for imaging flow in the fetal cardiovascular system, but these approaches have mostly been limited to single-slice acquisition[18–21]. 4D flow MRI has been applied in the sheep fetus[22] and pregnant rhesus macaques[23], while using anesthetic to immobilize the mother and fetus. A recent report showed 4D flow MRI results in the late-gestation human fetus[24] using an MRI-compatible Doppler ultrasound device[25,26] to cardiac gate cine images acquired during periods of low fetal motion. Volumetric imaging of the fetal heart is highly advantageous because it permits visualization of the intricate vasculature and the complex connections and shunts within the developing heart. Three-dimensional (3D) MRI of the human fetal heart can be performed[27] to provide a static visualization of external cardiac anatomy. However, as the heart is a highly dynamic organ, 3D imaging does not provide information about cardiac function or blood flow.

Anesthesia is normally precluded for human fetal imaging and the existing prenatal PC-MRI methods can be heavily compromised in the event of fetal motion. Minor fetal motion, such as small displacements compared to the target vessel, can compromise the accuracy of flow measurements, and at worst, for 2D imaging major fetal motion can shift the vessel out of the imaging plane entirely. Without motion-correction, these effects can lead to a scenario where the operator must repeatedly perform new, manually specified pilot scans, then reacquire the PC-MRI scan until the desired images are obtained. Reacquiring failed scans can be time consuming and challenging given limited available examination time and the size of the anatomy. Moreover, even when successfully achieved, two-dimensional imaging is intrinsically limited given the size of the heart and vasculature and wide range of anatomical arrangements in congenital heart disease.

Recently, we created an MRI framework for motion-tolerant 3D imaging of the fetal cardiovascular system based upon the imaging principle of acquiring multi-planar stacks of slices for volumetric reconstruction with slice-to-volume registration (SVR), originally developed for fetal brain imaging[28–31] and have demonstrated its clinical utility[27]. This was extended to time-resolved 4D cine imaging of the fetal heart[32,33] using a highly accelerated dynamic $k$–$t$ SENSE[34,35] balanced steady-state free precession (bSSFP) multi-planar acquisition combined with retrospective image-domain techniques for motion correction and cardiac gating. These approaches dispense with the need for precise single-slice planning; instead the operator is simply required to cover the fetal heart with a sufficient number of non-coplanar stacks. The final reconstructed 4D cine volumes allow for interpretation of the complicated fetal cardiac vessel structures and intra-chamber connections, in any 2D plane.

While 4D flow MRI is conventionally carried out in adults using spoiled gradient echo (SPGR) sequences[9,36,37], velocity-encoding can be achieved by exploiting the inherent flow sensitivity of bSSFP images[38–40]. The gain in signal-to-noise ratio (SNR) provided by bSSFP compared to SPGR is highly beneficial for fetal imaging where SNR is a major challenge because the fetus lies far from the radiofrequency (RF) receiver coils. bSSFP imaging has additional technical considerations[41,42], such as off-resonance effects, flow-related artefacts close to black bands, and higher specific absorption rate (SAR). However, the in utero environment (especially at ≤1.5 T field strength) is particularly amenable to bSSFP acquisitions as the uterus and the fetal lungs are filled with fluid, greatly mitigating the occurrence of banding artefacts. In addition, flow rates are reduced within the fetal heart compared to adults, reducing the effects of flow-related artefacts.

In this paper, the 4D cine framework is extended to generate in utero, motion-corrected, 4D flow volumes which depict temporally dynamic three-directional velocity vector fields of blood flow in the whole fetal heart and great vessels. As bSSFP sequences possess an intrinsic velocity sensitivity the diversely oriented stacks inherent to SVR methods provide multiple non-colinear sensitizations which can be used to recover full velocity information. Here, we explore this concept and present an initial framework for recovering fully motion-corrected 4D velocity vector fields of blood flow in the fetal heart. The velocity vector field reconstruction framework is validated in both simulated and physical flow phantoms. Finally, the full 4D flow cine framework is evaluated in a cohort of seven human fetal subjects: blood flow measurements are conducted by two blinded expert fetal cardiologists in five major vessels using the 4D flow cine volumes. With further development, this method could enable new clinical applications of 4D flow MRI for assessment of congenital heart disease[38–42].

## Results

**Validation of flow measurements in phantoms.** Results from the simulated flow phantom are shown in Fig. 1. The reconstructed velocity volume closely matched the original synthetic flow phantom. The difference image between the reconstructed velocity volume and the synthetic flow phantom (Fig. 1d) in regions corresponding to the pipes was within the level of the noise, except for regions which were not intersected by all five phase stacks (black arrow on difference image). This vulnerability to insufficient sampling motivated the use of five stacks for the in vivo data.

A 3D render of the plastic tube within the physical flow phantom is shown in Fig. 2a and a cross-sectional plane from the phantom is shown in Fig. 2b. Figure 2c shows velocity component and velocity magnitude maps reconstructed using a standard PC-SPGR method and the proposed multi-stack bSSFP method. The directionality and the speed of the flowing water was consistent between the bSSFP and the PC-SPGR velocity volume reconstructions. There was a small bias in the velocity values between the two methods. Measured across all voxels within the tube, the biases for the different velocity component volumes were: $V_x^{\text{bias}} = 4\%$,

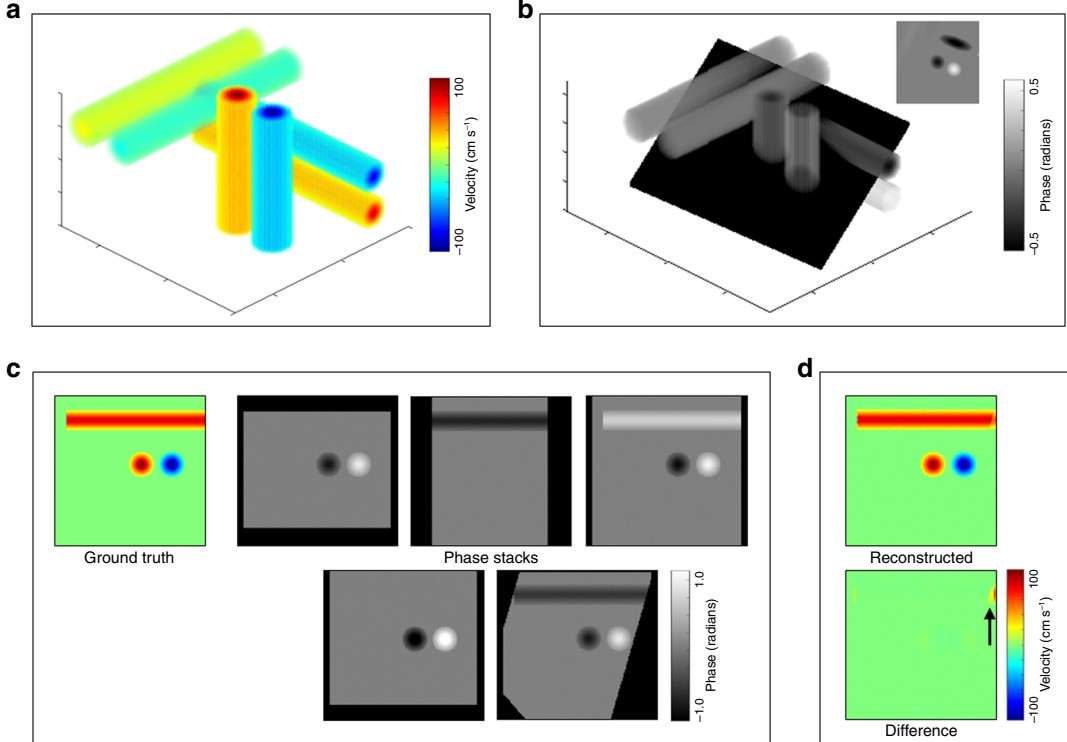

**Fig. 1 Velocity encoding with multi-planar stacks in a simulated flow phantom. a** The simulated flow phantom consisted of six orthogonal pipes with variable flow velocities ($\pm 25$, $\pm 80$, $\pm 100$ cm s$^{-1}$). **b** Example of a possible oblique slice projection (black plane) through the phantom and, inset, the corresponding in-plane phase image. **c** Ground truth flow velocities (color) and phase images (greyscale) from five non-coplanar stacks. The planes displayed are all positioned at the same spatial location. The phase values in the pipes change depending on the orientation of the stack. **d** Three-directional velocity reconstruction at an equivalent location to **c** and the corresponding subtraction image to show reconstruction errors. Black arrow shows region not covered by all phase stacks.

$V_y{}^{\text{bias}} = 3\%$, $V_z{}^{\text{bias}} = 1\%$, and for the velocity magnitude volume: $|\mathbf{V}|^{\text{bias}} = 4\%$. The associated root mean square errors (normalized to the 99 percentile range) were $\mathbf{V_x}^{n\text{RMSE}} = 23\%$, $\mathbf{V_y}^{n\text{RMSE}} = 21\%$, $\mathbf{V_z}^{n\text{RMSE}} = 19\%$, and $|\mathbf{V}|^{n\text{RMSE}} = 17\%$. The mean flow velocity measured with PC-SPGR was $4.0 \pm 1.4$ cm s$^{-1}$ and with bSSFP was $4.1 \pm 1.3$ cm s$^{-1}$.

Figure 2d shows flow measurements calculated in 20 regions of interest drawn perpendicular to the direction of flow at random locations throughout the flow phantom. For validation of the MRI flow measurements, the volume flow rate of the water was also determined using a measuring cylinder (Fig. 2d, red dashed line). The mean flow measurements using PC-SPGR, bSSFP, and the cylinder were $Q_{\text{PC-SPGR}} = 0.69 \pm 0.11$ ml s$^{-1}$, $Q_{\text{bSSFP}} = 0.72 \pm 0.11$ ml s$^{-1}$, and $Q_{\text{cyl}} = 0.74$ ml s$^{-1}$, respectively. There was a small bias of $-4\%$ between the PC-SPGR and bSSFP measurements and the standard deviation of the bias was 10%.

**Fetal whole heart 4D blood flow cine visualization.** The proposed 4D flow reconstruction framework produced coherent flow patterns in all fetal cases. For evaluation of this data, the 4D magnitude cine volumes were used to orientate the heart in any arbitrary view enabling both through-plane and in-plane examination of blood flow through specific vessels or regions of the heart. There is a large amount of data in each subject, so one exemplar normal subject (ID03, GA24$^{+2}$) will be described in detail, with additional references to an older, right aortic arch subject (ID06, GA32$^{+3}$). Supplementary Fig. 1 shows three different oblique cardiac views during systole and velocity-component volumes from the 4D flow cine reconstruction. In Supplementary Fig. 1a, the directionality of measured blood flow changes from positive to negative as the blood passes superiorly

via the ascending aorta before continuing in an inferior direction via the descending aorta. A horizontal component of blood flow can be seen at the transverse arch of the aorta. In Supplementary Fig. 1b, the components of slow blood flow in the superior vena cava can be seen, alongside the faster flowing ascending aorta and the pulmonary artery. In the third velocity–component volume ($\mathbf{V_3}$) of Supplementary Fig. 1c, blood in the aorta and pulmonary artery can be seen flowing with the same directionality.

4D flow cine volumes were generated by vector combination of the reconstructed velocity component volumes. Both 2D (Fig. 3) and 3D visualization (Fig. 4 and Fig. 5) of velocity vectors reveal patterns of blood flow consistent with expected fetal hemodynamics[43]. Pulsatile blood flow through the full extent of the aorta could be visualized through the course of the cardiac cycle (Supplementary Movies 1 and 4). During systole, high velocity blood flow is observed in the ascending aorta and descending aorta (Figs. 3a, 4a, and 5a). When orientated to visualize the outflow tracts, (Figs. 3b, 4b, and 5b, Supplementary Movies 2 and 5) simultaneous outflow from the left- and right- ventricles can be observed. 3D visualization in particular (Figs. 4b and 5b) demonstrates flow through the pulmonary artery and ductus arteriosus, and simultaneously the full path of aortic blood flow through the ascending aorta and then continuing to the descending aorta. Figures 3c and 4c show right atrial inflow from the superior vena cava and inferior vena cava. In Fig. 4c and Supplementary Movie 3, this blood flow can be seen exiting the ventricle via the pulmonary artery. Interatrial blood flow through the foramen ovale can also be seen in Fig. 3d.

Developmental changes in both cardiac structure and blood flow can be seen when comparing 4D flow cine volumes at different gestational ages (GAs). The fetal heart in Fig. 4 (ID03, 24$^{+2}$ weeks GA) is approximately 8-weeks younger than the heart in Fig. 5

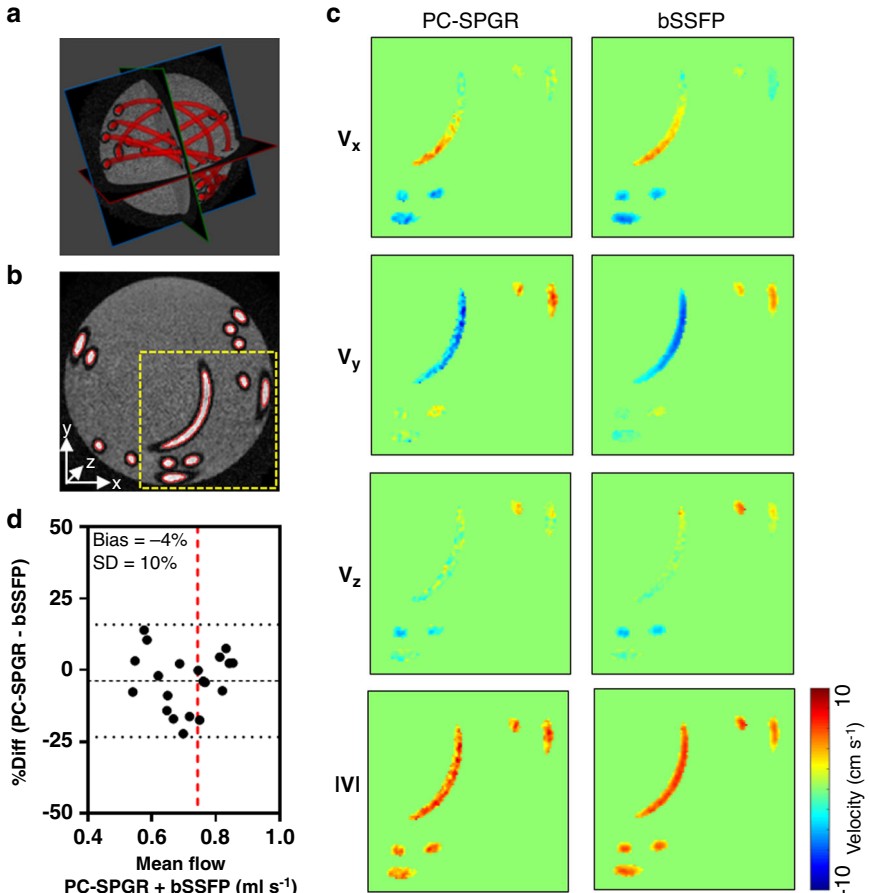

**Fig. 2 Validation of multi-planar velocity encoding in a physical flow phantom.** The physical flow phantom consisted of a spherical, water-filled glass flask containing a fixed-diameter tube connected to a flow pump. **a** 3D render showing the segmented plastic tube in red. **b** Cross-sectional magnitude image where the dashed-yellow line corresponds to the velocity maps in part **c** of the figure. **c** Measured velocity components $V_x$, $V_y$, $V_z$, and magnitude of the velocity $|V|$. Left column shows velocity components from the PC-SPGR acquisition. Right column shows velocity components from the bSSFP acquisition reconstructed using the proposed multi-stack velocity-encoding scheme. The mean (±SD) flow velocity across all voxels within the tube were $4.0 \pm 1.4\,\mathrm{cm\,s^{-1}}$ (PC-SPGR) and $4.1 \pm 1.3\,\mathrm{cm\,s^{-1}}$ (bSSFP). **d** Bland–Altman plot comparing mean flow values from ROIs ($n = 20$ independent regions) within the flow phantom (Bias = −4, SD of Bias = 10%). The red dashed line indicates the volume flow rate determined with a measuring cylinder. Source data are provided as a Source Data file.

(ID06, $32^{+3}$ weeks GA). The volume of the blood pool is much greater at $32^{+3}$ weeks, hence the velocity vector field has more points throughout the heart. The velocity vectors broadly have greater magnitude throughout the heart and vessels. Blood flow through the ductus arteriosus (DA) is clearly observed in the fetus at $32^{+3}$ weeks (Fig. 5a), whereas this flow is more difficult to distinguish in the smaller $24^{+2}$ weeks heart (Fig. 4a).

**Measurement of blood flow rates in fetal vasculature.** Both expert readers were able to successfully delineate the blood vessels in the 4D magnitude volumes in all but one vessel in the youngest subject (ductus arteriosus, ID03, $24^{+2}$ weeks GA). This was equivalent to a 4% region of interest (ROI) failure rate (Table 1). Across all trials, the ductus arteriosus was the most challenging vessel to segment with the lowest mean vessel delineation score (from 1.0 to 1.3, Table 1). The mean vessel delineation score was ≥2.0 for all other vessels. Flow curves generated using these ROIs (Fig. 6a, b) demonstrated pulsatile, phasic blood flow in the major arteries (the aorta, pulmonary artery, arterial duct, and descending aorta) in the majority of cases. The arterial flow curve failure rate was 3% (Table 1). In a small number of cases, the arterial flow curves were unexpectedly flat throughout the cardiac

cycle and did not demonstrate phasic behavior. Across all trials, the ductus arteriosus flow curves were the lowest scoring (from 1.8 to 2.4, Table 1) reflecting the difficulty in accurately delineating this anatomical region. In all cases, the superior vena cava had a more constant flow curve throughout the cardiac cycle, in keeping with venous return flow.

When taking the mean of the entire cohort (Fig. 6c), blood flow values were lower compared to measurements using a different MRI technique in term infants (i.e., older GA) by Prsa et al.[44] by 26–61% depending on the vessel. The pulmonary artery had the largest mean flow rate ($170 \pm 53\,\mathrm{ml\,min^{-1}\,kg^{-1}}$), followed by the ascending aorta ($141 \pm 57\,\mathrm{ml\,min^{-1}\,kg^{-1}}$), descending aorta ($127 \pm 33\,\mathrm{ml\,min^{-1}\,kg^{-1}}$), superior vena cava ($76 \pm 22\,\mathrm{ml\,min^{-1}\,kg^{-1}}$), and ductus arteriosus ($73 \pm 32\,\mathrm{ml\,min^{-1}\,kg^{-1}}$). Flow curves generated from the ROIs drawn by both expert readers had low intra-repeatability bias scores of 6% and 2%, respectively for Reader 1 and Reader 2 (Fig. 6d, e). The standard deviations of these biases were comparable between readers: Reader 1 = 23% and Reader 2 = 21%. The inter-reader bias (Fig. 6f) was larger at −13% indicating that measured vessel flows were slightly faster on average when segmentation was performed by Reader 2. The standard deviation of the inter-reader bias (28%) was larger than the intra-reader results.

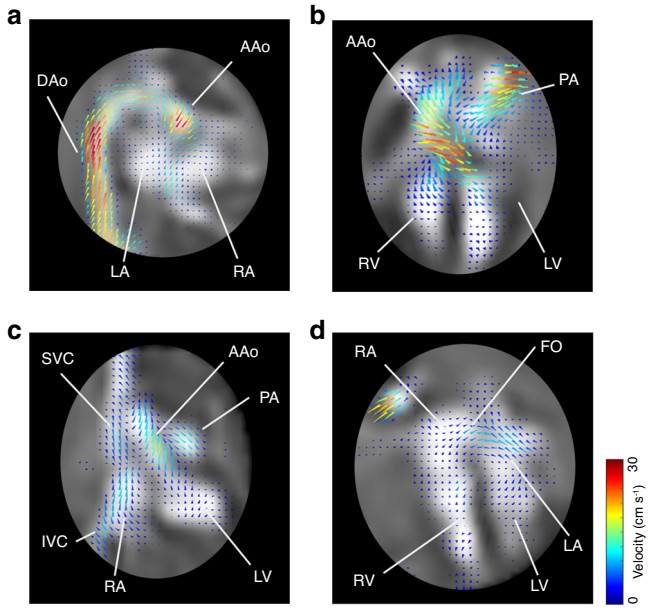

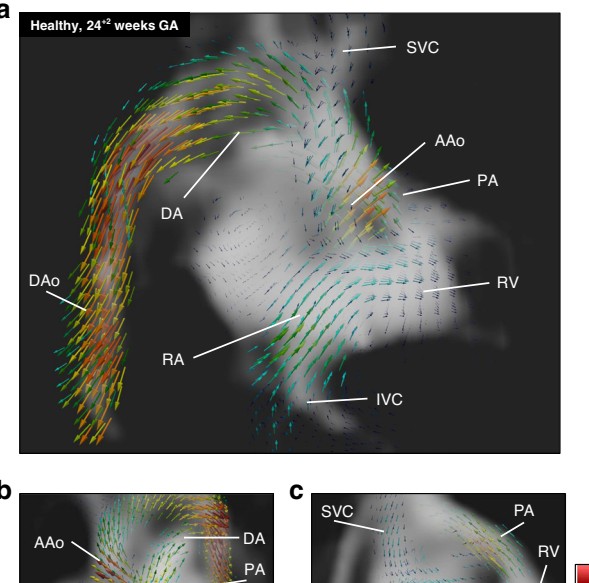

**Fig. 3 Cross-sectional visualization of 4D flow cine volumes.** Velocity vector plots in subject ID03 (healthy fetus 24$^{+2}$ weeks gestational age) overlaid on corresponding 4D magnitude cine frames. **a** Aortic arch plane showing blood flow vectors curve around the transverse arch of the aorta and down the descending aorta (DAo). **b** Dual-ventricle view showing blood flow vectors emerging from the left ventricle (LV) and right ventricle (RV) and passing through the ascending aorta (AAo) and pulmonary artery (PA), respectively. **c** Four-vessel view showing vectors flowing from the superior vena cava (SVC) and inferior vena cava (IVC) entering the right atrium (RA), and vectors passing through the AAo and PA. **d** Four-chamber view showing vector blood flow pass between the atria across the foramen ovale (FO).

**Fig. 4 Whole heart 4D blood flow in a healthy fetus.** Volumetric 4D flow renders showing three-directional flow velocity vectors in subject ID03 (healthy fetus 24$^{+2}$ weeks gestational age) overlaid on 4D magnitude cine renders, during end-ventricular systole. **a** Aortic arch view showing blood flow vectors emerging from the ascending aorta (AAo), flowing around the arch and through the descending aorta (DAo). **b** Long-axis view showing blood flow through the crossing AAo and pulmonary artery (PA). **c** Right-sided view showing right atrium (RA) inflow from the superior vena cava (SVC) and inferior vena cava (IVC), which subsequently mixes and passes through to the right ventricle (RV) before flowing out of the pulmonary artery (PA). See Supplementary information for movies of the views in this figure.

## Discussion

In this work, we present a method for in utero human whole heart fetal 4D cine quantitative blood flow imaging. 4D flow cine MRI reconstruction was achieved by exploiting the velocity-sensitive information inherent to the phase of dynamic bSSFP acquisitions in combination with SVR. Multiple, non-coplanar bSSFP stacks were used to reconstruct spatially identical, temporally resolved, motion-corrected magnitude, and vector flow volumes. Blood flow through fetal vasculature, the cardiac chambers and any other regions of interest could be inspected by reorientation of 3D renders into any view or re-slicing volumes into any desired 2D projection. The method was validated using a simulation and directly compared with conventional PC-SPGR quantitative flow measurements using a simple constant flow phantom consisting of coiled tube. Directionality of flows and absolute values measured using the proposed bSSFP method were found to be correct in both the digital and physical phantoms. Blood flow results obtained in the fetal subjects underestimated 2D PC-MRI measurements in normal term fetuses[44], however, the left and right outflow tracts demonstrated fast, pulsatile flow while the superior vena cava demonstrated slower, more uniform flow consistent with venous return. In this initial implementation of the framework, there are numerous potential contributing factors including limited spatial and temporal resolution, as well as possible errors in background phase and velocity corrections, but there are many opportunities for improvement and refinement.

One of the primary challenges of this work was reconstructing velocity measurements using PC MRI without maintaining a fixed imaging volume between different velocity-encoding acquisitions.

Volumetric reconstruction using SVR techniques[27–31] requires acquisition of non-coplanar stacks of images, therefore a novel approach to velocity-encoding was developed. Previous work demonstrated that quantitative flow imaging was feasible using phase images from bSSFP sequences[38]. Nielsen et al.[39] showed that multi-dimensional velocity-encoding could be achieved using balanced sequences by manipulating the gradient waveforms to impart orthogonal velocity-encoding, a concept which served as inspiration to combine multiple, unique velocity-encoding samples associated with different multi-planar stacks. However, rather than manipulating the gradient waveforms to sample velocity-space, instead, identical gradient waveforms were used for all the acquired image stacks and three-directional velocity-encoding was achieved purely through appropriate reorientation of the imaging volumes and the velocity sensitive gradient moments associated with these.

Here, all experiments were performed using a minimum of five image stacks. In theory, three-directional velocity encoding could be achieved using three stacks, however, fetal motion could then lead to local gaps in the sampled data that would compromise the mathematical inversion from component phases to full velocity, leading to inaccurate estimation of the final velocity vectors. Therefore, for this first demonstration of the proposed method we always required at least five image stacks with different orientations to ensure sufficient redundancy in the event of such motion. So far no attempt has been made to assess what is practically

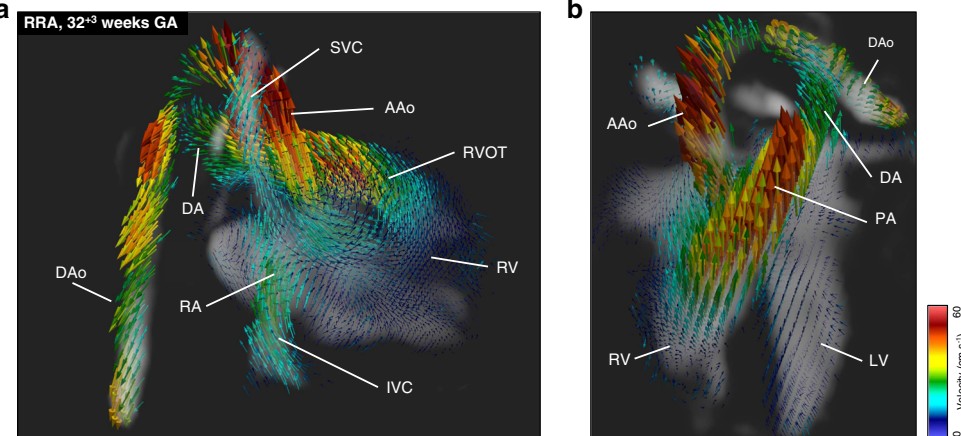

**Fig. 5 Whole heart 4D blood flow in a fetus diagnosed with congenital heart disease.** Volumetric 4D flow renders showing three-directional flow velocity vectors in subject ID06 (right aortic arch, fetus $32^{+3}$ weeks gestational age) overlaid on 4D magnitude cine renders, during end-ventricular systole. **a** Aortic arch view showing blood flow vectors emerging from the ascending aorta (AAo), flowing around the arch and through the descending aorta (DAo). Flow through the ductus arteriosus (DA) is clearly visualized. **b** Long-axis view showing blood flow through the crossing AAo and pulmonary artery (PA). At this developmental stage the blood flow velocity vectors are considerably faster compared to the $24^{+2}$ weeks gestational age fetus in Fig. 4. The velocity scalebar is adjusted to reflect this. See supplementary information for movies of the views in this figure.

| Table 1 Expert reader scores of vessel delineation and assessment of phasic behavior in arterial flow curves. | | | | |
|---|---|---|---|---|
| | **Vessel delineation score [1–3]** | | **Arterial flow curve score [1–3]** | |
| *Trial 1* | | | | |
| Vessel | Reader 1 | Reader 2 | Reader 1 | Reader 2 |
| Ao | 2.9 | 2.4 | 3.0 | 2.8 |
| DAo | 2.7 | 2.7 | 2.6 | 2.2 |
| PA | 2.6 | 2.0 | 2.7 | 2.4 |
| SVC | 2.7 | 2.9 | – | – |
| DA | 1.3 | 1.1 | 2.2 | 1.8 |
| *Trial2* | | | | |
| Ao | 2.7 | 2.6 | 2.9 | 2.7 |
| DAo | 2.6 | 2.7 | 2.7 | 2.1 |
| PA | 2.4 | 2.0 | 2.7 | 2.4 |
| SVC | 2.7 | 2.6 | – | – |
| DA | 1.1 | 1.0 | 2.4 | 1.8 |
| | ROI failure rate = 3% | | Flow curve failure rate = 4% | |

Scores are shown for Reader 1 and Reader 2, with 3- and 5-years of experience reading cardiac MRI, respectively. Values listed are the mean scores from all subjects ($n = 7$) of qualitative observations rated on 1-3 scoring scales. Note, SVC flow curves were not assessed as this venous vessel exhibits minimal phasic behavior. ROI failure rate represents the total number of vessels (across all trials, $n = 140$), which were not segmented by the expert readers. Flow curve failure rate represents the total number of arterial flow curves (across all successfully delineated arterial vessels, $n = 108$), where phasic variation was not observed throughout the cardiac cycle.

required of the imaging stacks, or what mitigation is appropriate to avoid incorrect results.

The physical flow phantom, which consisted of a plastic tube of flowing water submerged in a glass flask, demonstrated that the principles of non-coplanar velocity-encoding worked in practice. The flow measurements using the bSSFP acquisition scheme were accurate and directionally aligned with a gold-standard PC-SPGR acquisition. Both of the PC-SPGR and bSSFP mean flow measurements were consistent with the flow rate determined using a measuring cylinder. There was a small amount of variation in the flow values measured across the 20 ROIs and there was a small bias measured between the two MRI methods when comparing all voxels within the pipe. These variations were attributable to experimental imperfections such as registration offsets, partial volume effects, 2D ROI placement and positional drift of the pipes during scanning.

For this exploratory work, a total of seven fetal cases were reconstructed including three healthy subjects, two right-sided aortic arch cases and two further abnormal subjects, ranging from 24- to 32-weeks GA (mean GA = $30 \pm 2.8$ weeks). Drawing on previous clinical experience, we expected the vessel flow rates to be comparable between these particular subjects despite the variation in clinical status, although we acknowledge this is an assumption of the study. 4D flow cine volumes were successfully reconstructed in all subjects, with pulsatile blood flow measured in all subjects. Only the ductus arteriosus in the youngest subject (ID03, 24 weeks GA) could not be reliably delineated by the expert readers, illustrating the excellent image quality of the 4D reconstructions. Analysis of the arterial flow curves demonstrated that 97% of the flow curves measured demonstrated phasic behavior through the cardiac cycle. Consistent velocity magnitudes (ranging between 0 and an upper limit of 90 cm s$^{-1}$) were observed between reconstructions, with fast blood flow through the aorta and pulmonary artery and slower blood flow in the inferior vena cava, superior vena cava, and the chambers of the hearts.

The background phase correction of the bSSFP phase images, demonstrated by Nielsen et al.[39] and implemented here, was imperative for accurate inversion and calculation of velocity vectors. There are many system-related factors which can contribute phase offsets including eddy currents[45], concomitant gradients[46], and nonuniform imaging gradients[47], however, the stringent PNS and acoustic noise constraints required for fetal imaging mean that the imaging gradients are relatively benign compared to adult imaging. These effects and any other slowly varying phase changes, such as due to maternal motion, were greatly reduced and subsequently eliminated by the background phase correction. In addition, the uterus is particularly amendable to this method of phase correction because amniotic fluid fills the fetal lungs and surrounds the fetal body; this helps to minimize bSSFP-related artefacts, which would be problematic in the lungs of adults due to air, as well as in neonatal and pediatric subjects. Imaging at a relatively low 1.5 T-field strength helps to reduce phase variation as well. The proposed framework is field-strength independent, but the bSSFP acquisition will need optimization and testing at different field strengths.

Validation of fetal cardiac blood flow measurements is very challenging for any new method as the existing clinical standard,

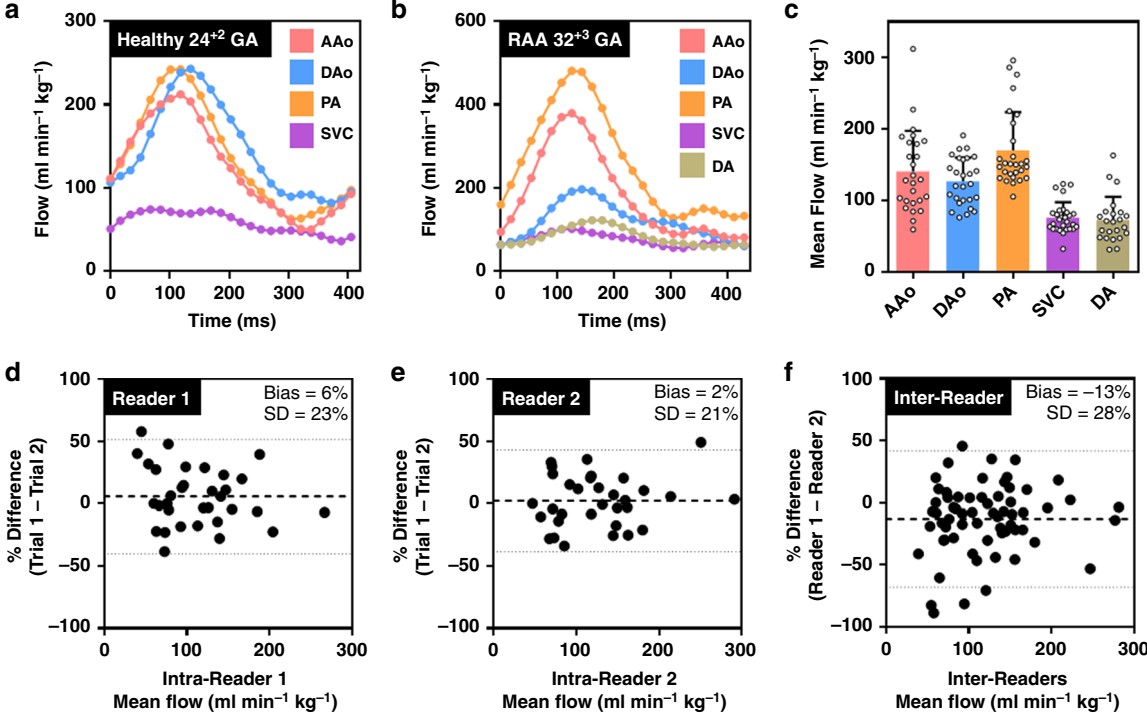

**Fig. 6 Analysis of fetal blood flow through the major vessels.** Flow curves measured in **a** subject ID03 (healthy fetus, 24$^{+2}$ weeks GA) and **b** subject ID06 (right aortic arch, 32$^{+3}$ weeks GA). The flow curves were derived from the 4D flow reconstructed volumes by transferring 2D cross-sectional ROIs drawn in planes perpendicular to the local vessel axis on the 4D magnitude volume reconstructions by expert fetal cardiologists. Note, that for subject ID03 in panel **a** the DA could not be reliably delineated. **c** Mean blood flows in five vessels across all seven fetal subjects measured twice by each of the two expert readers. Values are lower than previously published values (26–61% depending on vessel) in term fetuses[44]. AAo ascending aorta, DAo descending aorta, PA main pulmonary artery, SVC superior vena cava, DA ductus arteriosus. Bland–Altman plots showing **d**, **e** intra-reader and **f** inter-reader repeatability of mean flow values. Inter-reader bias and SD was slightly larger than the intra-reader bias for both experts. In **c**, data are represented as mean + SD. Individual points depict ROIs drawn by blinded expert readers across both trials ($n_{AAo}$ = 26, $n_{DAo}$ = 27, $n_{PA}$ 28, $n_{SVC}$ 28, $n_{DA}$ = 23. Individual measures were omitted from mean flow calculations if the vessel was not successfully delineated or the arterial flow curve showed no phasic variation). In **d**–**f**, dots depict paired measurements (intra-reader 1, $n$ = 33, intra-reader 2: $n$ = 32, inter-readers: $n$ = 64). Source data are provided as a Source Data file.

Doppler ultrasound, has various technical limitations (outlined in the Section "Introduction"[4–8]) and is highly dependent on the user and the fetal lie, which cause measurement uncertainty. This means that a direct comparison with Doppler ultrasound requires many subjects to be studied to draw fair and statistically meaningful conclusions. Comparison with other methods including ultrasound will be the subject of future clinical studies for which it will be necessary to accumulate sufficient subject numbers, but for this technical study quantification of blood flow curves in the major vessels was carried out in keeping with current PC-MRI blood flow methods[18,19], using 2D cross-sectional regions of interest defined on anatomical scans and transferred onto velocity maps. In comparison with term fetuses (mean GA = 37 ± 1.1 weeks) imaged using single-slice metric-optimized gating (MOG) fetal blood flow techniques[44], the mean absolute blood flow values measured here were lower by between 26% (ascending aorta) and 61% (ductus arteriosus) depending on the vessel. The relative absolute flow rates were broadly in keeping with literature values[44]: flow was fastest in the outflow tracts and slower in the inferior vena cava and superior vena cava, all of which could clearly be observed in the 4D vector flow visualizations. The blood flow measurements in the ductus arteriosus were considerably lower than literature values, which tallied with the expert reader vessel delineation scores and phasic flow curve scores, where it was scored markedly lower than the other vessels.

The overall directionality of the vectors within the 4D flow cine volumes, phasic behavior through the cardiac cycle, and faster

velocity vectors in older fetuses all provide confidence that the underlying principles of our framework are correct. However, some degree of intra-subject variability was observed in some small localized regions, where velocity vectors pointed in unexpected orientations, such as perpendicular to the anticipated direction of flow. A range of issues could explain this including incomplete velocity-encoding due to fetal and cardiac motion, phase changes due to local fetal movement not sufficiently compensated for by the velocity drift correction, or regions not sampled by a sufficient fraction of the phase stacks.

The low absolute values of the blood flow curves can be partly attributed to the limitations of the implemented bSSFP acquisition. Previous studies have shown that $k$–$t$ accelerated 4D flow MRI acquisitions in adults underestimate peak flow[48–50], however, a more significant limiting factor for our fetal study is probably the relatively low spatial resolution (2.0 × 2.0 × 6.0 mm acquisition interpolated to 1.25 × 1.25 × 1.25 mm using super-resolution reconstruction) and relatively long temporal resolution (72 ms acquisition interpolated to 25 frames) compared to the sizes of the fetal blood vessels and the fetal heart rate. The fetal outflow tracts have previously been measured with diameters ranging from ~4 to 5 mm across the aortic valve, ~5 to 6 mm across the pulmonary valve, and ~2.5–3.5 mm across the arterial duct (24–36 weeks GA)[51]. These diameters are equivalent to between three and seven voxels (given partial volume effects) in our volumetric reconstructions, which can lead to blurred boundaries in the smallest vessels, as in the ductus arteriosus of

the youngest fetus (ID03: $24^{+2}$ weeks GA, Fig. 4a). The fetal cardiac cycle has a typical duration between 330 and 540 ms meaning that the implemented bSSFP acquisition collects ~5–7 frames per heartbeat. By comparison, cine imaging in adult hearts, where the cardiac cycle is longer, can typically be configured to collect >30 frames per cardiac cycle. The relatively small number of frames acquired through the fetal cardiac cycle with the proposed framework will cause temporal averaging of measured flow values. This limited spatial and temporal resolution is a priority technical challenge to address.

In general, the limitations of spatial and temporal resolution as well as fetal motion are the most challenging aspects of fetal flow MRI. The current standard for 2D fetal flow MRI using MOG[44] has a spatial resolution of $1.25 \times 1.25 \times 5.0$ mm and temporal resolution of 50 ms. Motion-correction is possible on a per slice basis but through-plane displacement of the fetus cannot be corrected and the data is rejected[19]. Preliminary experiments using a Doppler ultrasound-triggered device for 4D flow MRI[24] have a spatial resolution of $1.04 \times 1.04 \times 5.0$ mm and temporal resolution of approximately 40 ms. This method is very promising as it has the benefit of allowing MRI systems to perform standard cardiac-gated acquisitions that have already been developed for adults and children. However, if the fetus moves significantly, which is common especially in younger nonterm subjects, then motion-correction techniques will be necessary for accurate 4D flow measurements. Our proposed method performs image-based cardiac synchronization without the need for extra hardware, but if prospective triggering is advantageous or preferable for the user, the velocity-encoding principles and SVR motion-correction underpinning our method are fully compatible with prospectively-gated acquisitions.

An acquisition scheme specifically tailored to 4D flow cine imaging with improved spatial and temporal resolution will be the focus of future methods development work. Despite the present limitations, our acquisition protocol brings significant advantages compared to single-slice methods as it eliminates the need for high precision slice plane selection during planning, which is challenging and prone to error. The proposed method also allows for simultaneous structural imaging and complete flow mapping. This is clinically beneficial and more meaningful as it enables contemporaneous inspection of flow patterns at diverse locations throughout the fetal circulatory system. The potential for studying flow can be extended to intra-cardiac flow patterns, which could enhance understanding of perturbations in the setting of structural lesions.

In conclusion, a fully motion-corrected 4D vector flow cine imaging method for fetal cardiac visualization based on accelerated bSSFP sequences has been presented. The method was validated using simulations and a constant flow phantom before being deployed in seven fetal subjects. Results confirm that the method is robust, provides coherent temporally and spatially resolved vector velocity fields, and can yield quantitative flow data for major fetal vessels. At present, flow rates measured in fetal vessels are lower than would be expected but there are many opportunities for improving and refining the method and these will be the subject of future technical research. Even with this initial implementation, the framework enables simultaneous visualization of the anatomy and blood flow in the whole heart, allowing for comprehensive assessment of fetal circulation, using a motion-robust acquisition that avoids having to reacquire scans during an examination. The potential ability to visualize three-directional vascular flow patterns in prenatal life, in conjunction with 4D anatomical imaging, could offer exciting new opportunities for assessment of the fetal cardiovascular system in both health and disease.

## Methods

**Velocity-encoding with bSSFP**. When measuring a single-component of flow with a conventional PC MRI sequence, such as a SPGR, two measurements of phase are acquired with identical sequence parameters except for different velocity-encoding gradients. Subtraction of the two phase images removes background phase offsets and phase associated with stationary tissue, allowing for calculation of the projection of any arbitrary velocity vector on to a specified spatial direction. The magnitude and direction of velocity-sensitivity is determined by the first moment ($\mathbf{m}$ in Fig. 7) of the combined imaging and velocity-encoding gradients. For conventional three-directional PC flow imaging, the imaging geometry is held fixed and the velocity-encoding gradients, which are usually simple bipolar trapezoids with no zeroth order moment, are switched to different axes to create a set of orthogonal velocity-sensitive imaging volumes, as shown in Fig. 7a, b.

Adding dedicated bipolar velocity encoding gradients into bSSFP sequences is challenging as these increase the repetition time (TR). However, Markl et al.[9] devised a method for measuring through-plane flow using a bSSFP sequence by simply inverting the slice-select gradient between two consecutive acquisitions. As the readout gradients are identical between the two acquisitions, subtraction of the resulting phase images eliminates any contributions caused by all other imaging gradients. The resultant velocity-encoding is associated only with the slice-select gradient axis. Nielsen et al.[39] proposed a time-efficient method of three-directional velocity mapping using bSSFP by minimally augmenting the imaging gradients to create an additional controllable velocity sensitivity. Acquisition of three sequences with unique gradient first moments allowed for estimation of three-directional flow vectors.

For fetal imaging with SVR, acquisition of multi-planar stacks in different orientations is a prerequisite for volumetric reconstruction. SVR with conventional three-directional velocity-encoding would be prohibitively time-inefficient because a minimum of nine acquisitions are required to sample three spatial dimensions and three velocity directions. Instead, we propose achieving multi-dimensional velocity-encoding by rotating stacks with identical fixed gradient first moments, as shown in Fig. 7c, d. With this scheme, a minimum of three non-coplanar stacks with (ideally orthogonal) non-colinear gradient first moments are required to sample three spatial dimensions and three velocity directions.

In our previous framework for fetal whole heart 4D cine imaging[33], a bSSFP sequence was used for image acquisition. The framework could be applied to SPGR acquisitions, but bSSFP permits efficient simultaneous acquisition of both anatomical and flow information with strong SNR performance in both cases—SPGR scans may produce phase maps of sufficient quality, but the anatomical imaging would be compromised. All three of the gradients that form bSSFP sequences with Cartesian sampling generally contribute nonzero first moments, but only the readout and slice-select gradients produce constant effects throughout the acquisition. The net effect is an oblique gradient first moment. This is directed in a plane containing the slice select and readout directions, since the effect of the phase encoding gradient, which is not zero, averages to zero. If acquisitions with three orthogonal bSSFP planes are employed, the three associated gradient moments are also orthogonal to each other (Fig. 7d), but they are rotated relative to a standard three-directional velocity encoding scheme (Fig. 7b).

In this work, we take advantage of the diversity of imaging planes required for volumetric reconstruction of fetal hearts to sample multiple velocity-encoding directions and then to recover a full vector representation of blood flow using a scattered data SVR approach adapted from fetal diffusion MRI[52]. In theory, as in the studies by Nielsen et al.[39], a minimum of three stacks with non-colinear first moments are required for three-directional velocity-encoding, but to ensure stable inversion of phase data into velocity vectors given fetal motion the present experiments were performed with a minimum of five bSSFP stacks.

**Fetal whole heart 4D flow cine MRI framework**. The proposed framework for whole heart 4D flow cine reconstruction is shown in Fig. 8. In brief, the reconstruction consisted of two streams: first, motion corrected 4D magnitude cine reconstruction was performed adhering to the pipeline previously described by van Amerom et al.[33]. The data consisted of multiple stacks of highly accelerated dynamic bSSFP imaging planes, with multiple time frames obtained for each slice. The temporal resolution of the acquisition (72 ms per frame) was sufficient to capture fetal cardiac motion in real-time without the requirement for periodic movement and electrocardiography-gating, however, note that the $k$–$t$ SENSE acquisition itself was not interactive due to image reconstruction latency (on the order of minutes)[53]. The fetal heart rate was estimated from the frames of each acquired slice and used to assign cardiac phases to each successive frame. The cardiac phases of different slice locations were synchronized and then rigid body transformations were determined for each individual frame to provide a fully motion corrected dataset for reconstruction. An anonymised example dataset of subject ID03 is available from https://doi.org/10.6084/m9.figshare.c.4689437.

Parameters generated from this pipeline, including stack and frame spatial transformations, motion correction parameters, and cardiac synchronization parameters were then passed to the phase data reconstruction stream. Background phase subtraction, gradient moment reorientation, and calculation of velocity volumes were performed, before a final velocity correction was applied. The complete framework resulted in coupled 4D magnitude cine and 4D flow cine

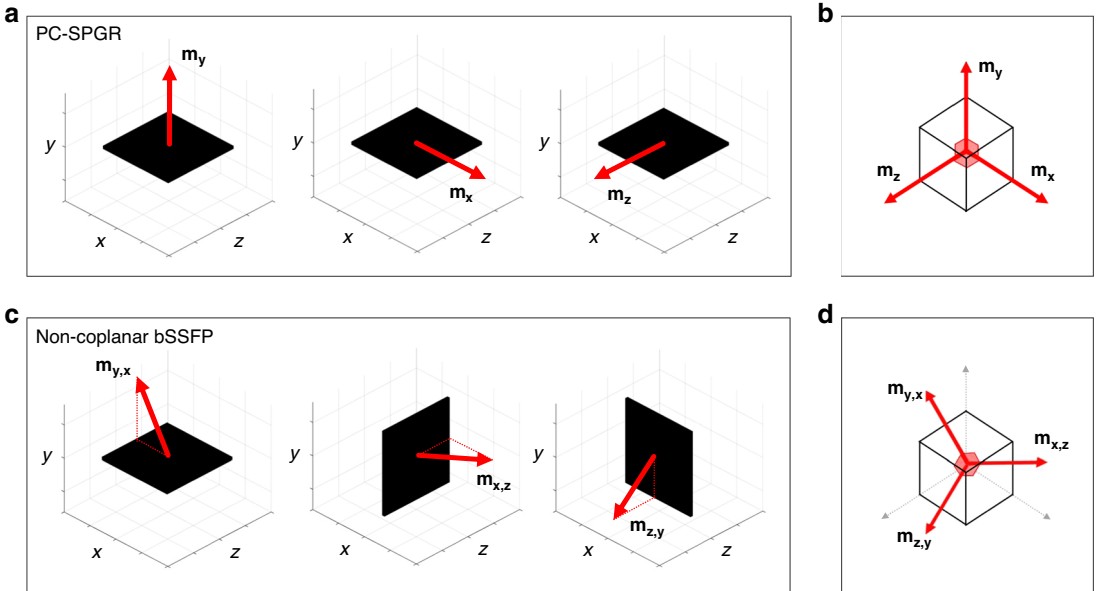

**Fig. 7 Velocity encoding using a standard paradigm versus a non-coplanar multi-stack approach compatible with slice-to-volume registration (SVR).**
**a** In standard velocity-encoding, such as using a spoiled gradient echo (SPGR) sequence, the imaging slice (black) is held fixed and additional dedicated velocity sensitizing gradients are added to create first moments (**m**) in specified spatial directions (**m**$_x$, **m**$_y$, **m**$_z$, where subscripts denote the spatial directions of the velocity encoding). Note that there are also typically first order moments associated with the native imaging gradients, which add a constant additional encoding, but this is not shown as it is usually subtracted out. **b** For each imaged voxel, the three velocity-encoding directions are orthogonal and conventionally aligned with the imaging axes. **c** Volumetric reconstruction using SVR requires the acquisition of non-coplanar stacks. In this case, the directionality of the first moment associated with the intrinsic imaging gradients is maintained relative to each acquired slice and this provides three-directional velocity-encoding. Slice selective imaging sequences generally have intrinsic velocity-sensitivity that is oblique to the slice plane. In the case of the bSSFP sequences deployed in this study senitization is along the slice and read directions (hence **m**$_{y,x}$, **m**$_{x,z}$, **m**$_{z,y}$ in the figure). Combining three non-coplanar stacks of slices (**d**) provides data at the location of each voxel with three non-colinear velocity sensitizations, now oblique to the imaging axes. (Note that in this figure the coordinate axes show spatial directions. The arrows indicate only the directions in space of the relevant gradient moments).

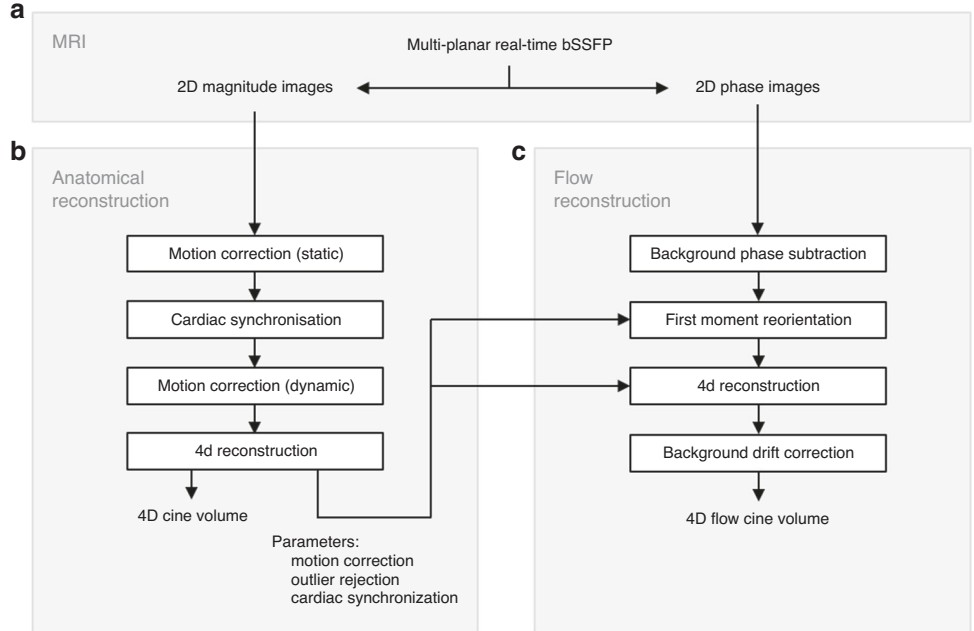

**Fig. 8 Framework for 4D flow cine volume reconstruction. a** Multi-planar dynamic bSSFP data is acquired and reconstructed into 2D magnitude and phase images using $k$–$t$ SENSE. **b** Anatomical, volumetric 4D magnitude cine reconstruction of the fetal heart is performed[33]. **c** Volumetric 4D vector flow cine reconstruction of the fetal heart is performed, using motion correction, cardiac synchronization, and frame outlier rejection parameters from the magnitude reconstruction pathway. Background phase is estimated and subtracted by polynomial approximation. First moment vectors for the imaging gradients, as specified for the native acquisitions, are rotated as a result of detected motion. Velocity vector fields are recovered by inversion of the phase image information using a conjugate gradient method. Volumetric and temporal 4D reconstruction is performed before a final background velocity drift correction is applied.

volumes which are spatially and temporally equivalent. For clarification, we use the terminology "velocity volumes" and "velocity component volumes" throughout this manuscript as the fundamental output of our reconstruction pipeline is three 4D volumes of scalar values corresponding to the Cartesian components of the reconstructed velocity vector field, i.e., $V_x$, $V_y$, $V_z$ at each location in 3D space and cardiac phase. We also refer to the "velocity magnitude volume", which is the element-wise square-root of the sum of squares of the velocity component volumes, i.e., $|V|$.

**Multi-planar MRI acquisition**. Multi-planar, dynamic MR images were acquired in stacks of parallel slices with 96 time frames per slice position and reconstructed using $k$–$t$ SENSE[34] (acceleration factor 8). Stacks were acquired in multiple orientations to ensure full coverage of the heart and the great vessels. The acquired data form a set of $N_k$ MR image time frames of complex type, $\tilde{Y} = \{\tilde{Y}_k\}_{k=1,\ldots,N_k}$, where each frame, $k$, has an acquisition time $t_k$ and consists of elements $\tilde{y}_{jk}$ at 2D spatial coordinates indexed by $j$. The complex data was reconstructed into magnitude image frames, $Y = \{Y_k\}_{k=1,\ldots,N_k}$, and phase image frames $\Phi = \{\Phi_k\}_{k=1,\ldots,N_k}$.

**4D magnitude cine reconstruction**. The magnitude image frames, $Y$, were used for reconstruction of a 4D magnitude cine volume $X = \{X_h\}_{h=1,\ldots,N_h}$, where $X_h$ has elements $x_{ih}$ for spatial index $i$ and temporal index $h$ corresponding to cardiac phase, $\vartheta = \{\vartheta_h\}_{h=1,\ldots,N_h}$, as described in our previous framework[33]. The magnitude pipeline also includes an outlier detection process that operates at the voxel and frame level and is used to reduce the weight of discordant signals that are likely to be causes by image artefacts. The spatial locations of voxels in the magnitude images were identical to those in the phase images as both image types were generated from the same complex-valued acquisition. Therefore, reconstruction parameters and weights from the 4D magnitude cine stream could be passed to the 4D flow cine reconstruction stream. These included the following:

1. Rigid body transformation matrices, $A$, which align dynamic image time frames with the output cine volume.
2. Image frame outlier rejection weights, $p_k^{frame}$, which exclude or include the specified frames from further processing.
3. Cardiac cycle parameters, $t_l^{RR}$, which denote the duration of the R–R intervals for the slices, $l$, in the input stacks and, $\vartheta$, which map acquired image frames with phases of the cardiac cycle.

**Background phase correction**. Flow quantification with PC MRI relies on measuring changes in the phase signal due to blood flow. Phase measurements are known to be affected by hardware-related sources of phase offset[54] including eddy currents[45], concomitant gradients[46] and nonuniformities in the imaging gradients[47], however, these effects are inherently reduced in fetal scanning because the need to minimize acoustic noise and peripheral nerve stimulation (PNS) in the mother mean that the imaging gradients are relatively benign compared to adult PC-MRI. For prenatal MRI, fetal movement and maternal motion, such as respiration, are the primary sources of phase offset. All of these sources can combine to give a background phase offset which needs to be corrected in order to accurately estimate the velocity-induced phase changes within the heart and major vessels.

After some initial experimentation, we estimated the background phase offset of each stack by fitting a third-order polynomial[39] (Fig. 9d) to the phase images. A multi-planar ROI, which included the contents of the uterus whilst excluding the heart, was drawn on the magnitude images (Fig. 9a) and then transferred to the phase data to estimate the third-order polynomial. While the contents of the uterus are not necessarily static (such as the amniotic fluid or low-level fetal movements), the phase changes across this broad region are small and vary slowly, whereas the phase changes in the heart are greater and highly localized. Therefore, subtraction of the fitted polynomial resulted in corrected phase images proportional to the underlying velocity, with minimal remaining background phase variation (Fig. 9e).

**Gradient moment correction**. All slices within a single stack of phase data were acquired with the same configuration of gradient first moments, which corresponds to a fixed velocity-encoding direction relative to the slice. If fetal and/or maternal motion caused the slice to be rotated in anatomical space, then the gradient first moment was subject to the same rotation. However, the rotated slice is still a valid projection of the velocity given the change in gradient first moment. In the proposed framework, the gradient first moments of each frame were recalculated by applying the same frame transformation matrices ($A_k$) that were applied to the equivalent frames in the 4D magnitude cine reconstruction.

**4D flow cine reconstruction**. For a single bSSFP stack acquired with Cartesian sampling, the phase in a voxel, $\phi_{jk}$, is given by the dot product between the gradient first moment vector for the frame, $m_k = \{m_{q,k}\}_{q=1,2,3}$, and the local velocity vector, $v_{jk} = \{v_{q,jk}\}_{q=1,2,3}$, in voxel $j$ of the frame $k$, with $q$ indexing Cartesian components of the vectors:

$$\phi_{jk} = \gamma\left(m_{1,k}v_{1,jk} + m_{2,k}v_{2,jk} + m_{3,k}v_{3,jk}\right), \tag{1}$$

where $\gamma$ is the gyromagnetic ratio. In conventional velocity-encoding, $m_{q,k}$ would typically denote the readout, phase-encode and slice-select directions, but more generally $q$ represents any orthonormal coordinate system that is independent of the geometry of any particular slice stack. For scanner reconstructions, $q$ is in world coordinates. Note, that in a bSSFP sequence, the gradient first moment associated with the phase-encode direction is non-zero for each individual phase-encoded data line, but for the data as a whole there is no net moment.

Fetal and maternal motion causes rotation of the gradient first moments. The complete set of gradient first moments $M_k = \{m_k\}_{k=1\ldots N_k}$ for all frames, $k$, were corrected using the frame-wise transformations $A_k$ from the 4D cine magnitude reconstruction: $M_k^* = A_k M_k$, where $M_k^*$ are the corrected gradient first moments.

The final 4D cine velocity vector volume was defined as $V = \{V_h\}_{h=1\ldots N_h}$, where $V_h$ has vector elements $v_{ih} = \{v_{q,ih}\}_{q=1,2,3}$ for spatial index $i$ and temporal index $h$ corresponding to cardiac phases $\vartheta$. The MR image acquisition model describes the relationship between the acquired phase image frames, $\Phi_k$, the

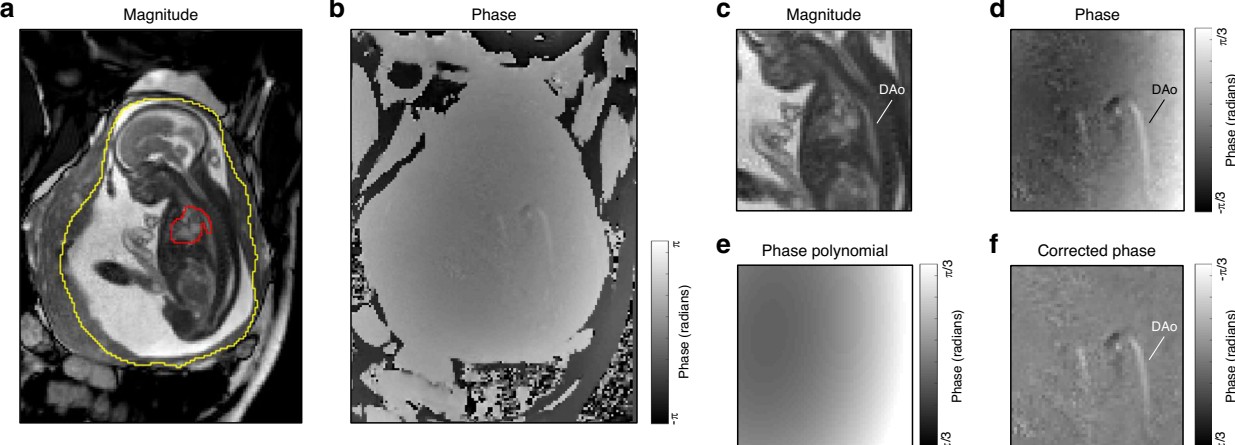

**Fig. 9 Background phase correction of bSSFP images.** Images shown were acquired in a healthy 24$^{+2}$ week fetus (ID03). **a** Magnitude image showing the in-plane regions of two volumes of interest drawn around the mother's uterus (yellow) and the fetal heart, including great vessels (red). **b** Corresponding uncorrected phase image. **c** Zoomed-in magnitude image showing the descending aorta (DAo). **d** Zoomed-in uncorrected phase image demonstrating the background phase variation, windowed to accentuate phase roll. **e** Third-order polynomial fit calculated from the uncorrected phase image using the uterus volume of interest but excluding the fetal heart and great vessels. **f** Corrected phase image after subtraction of the background phase estimate leaving localized phase changes that are then used to calculate the velocity vector field.

corrected gradient first moments $\mathbf{M}_k^*$ associated with frame, $k$, and the 4D cine velocity volume $\mathbf{V}$

$$\Phi_k = \gamma \sum_h \mathbf{W}^{hk} \mathbf{M}_k^* \mathbf{V}_h, \qquad (2)$$

where $\mathbf{W}^{hk}$ are the product of spatial and temporal weights taken from the 4D magnitude cine framework[33].

$\mathbf{V}$ was initialized equal to zero and then the error in each voxel was calculated between the acquired phase image voxels, $\phi_{jk}$, and simulated phase image voxels, $\hat{\phi}_{jk}$, as predicted using the acquisition model in Eq. (2):

$$e_{jk} = \phi_{jk} - \hat{\phi}_{jk} = \phi_{jk} - \gamma \sum_j \sum_k w_{ij}^{hk} \mathbf{m}_k^* \mathbf{v}_{ij}, \qquad (3)$$

$\mathbf{V}$ was estimated through minimization of the error term in Eq. (3) by calculation of the sum of squares difference using an iteratively optimized conjugate gradient descent method

$$\mathbf{V} = \underset{\mathbf{V}}{\text{argmin}} \left( \sum_j \sum_k \lVert e_{jk} \rVert^2 + \lambda \text{Reg}(\mathbf{V}) \right), \qquad (4)$$

which includes a spatial edge-preserving regularization term $\text{Reg}(\mathbf{V})$ to stabilize the reconstruction and a regularization controlling parameter, $\lambda$. The minimization was achieved by gradient descent using explicit differentiation of $\{e_{jk}\}^2$ with respect to each of the three Cartesian components $v_{q,jk}$.

**Velocity drift correction**. The initial background phase correction sremoved large length scale phase changes. However, following reconstruction, it was observed that some background regions of tissue in the 3D magnitude volumes still displayed a velocity offset in the 4D cine velocity–component volumes, although there was no signature for this readily observed in the individual phase images. To correct for this residual offset, the median velocity in non-blood pool regions for each velocity component volume, $\mathbf{V}_q = \left\{ v_{q,ih} \right\}_{q=1,2,3; i=1\ldots N_i; h=1\ldots N_h}$, was calculated over all frames, $h$, and used to create a third-order polynomial; this polynomial was then subtracted from $\mathbf{V}_q$, which resulted in a background drift-corrected velocity component volume. The mask of non-blood pool regions was generated by thresholding the magnitude cine volume ($\mathbf{X}$). Only spatial locations within $\mathbf{X}$ which did not contain blood in any frame of the cardiac cycle were classified as non-blood pool.

**Simulated flow phantom study**. A simulated flow phantom was created in MATLAB for the purposes of testing and validating the reconstruction of multi-planar stacks of phase data into velocity volumes. The simulated flow phantom consisted of six cylindrical pipes arranged in pairs with antiparallel flow, oriented in three orthogonal directions. Gaussian flow profiles, which were truncated at the boundaries of the cylinders, were used as a simple approximate of laminar flow in a pipe. The pairs of pipes had different peak velocities ($\pm 25$, $\pm 80$, and $\pm 100$ cm s$^{-1}$) chosen to simulate a range of different, physiologically relevant flow rates. Multi-planar stacks of bSSFP phase images in any orientation were simulated according to Eq. (1) (VENC = 159 cm s$^{-1}$). Gaussian random noise was added to the phase images so that they had a SNR equivalent to the bSSFP data acquired in the physical flow phantom experiment.

For the results presented, five stacks of images with 1 mm isotropic voxels were generated in different orientations: three stacks were chosen such that each was orthogonal to one set of pipes, and two stacks were specified in oblique orientations through the simulated flow phantom. Velocity volumes were then reconstructed using the proposed framework and the results were compared with the ground truth simulated flow phantom. The simulated flow phantom MATLAB code can be found online at https://github.com/tomaroberts/synthflow_phantom.

**Physical flow phantom study**. A simple physical constant flow phantom, which consisted of a long, fixed-diameter plastic tube connected to a water pump, was scanned to demonstrate the proposed velocity reconstruction method. To minimize bSSFP-related artefacts, the plastic tube was submerged in a water-filled, spherical glass flask.

Five bSSFP stacks were acquired in the physical flow phantom: three in orthogonal orientations aligned with the scanner axes and two oblique stacks. Images were acquired with $1.25 \times 1.25$ mm in-plane resolution and 1.25 mm slice-thickness. The FOV was 150 mm$^3$. With these parameters the velocity that produced a $\pi$ phase shift, VENC$_{bssfp}$ was 30 cm s$^{-1}$. This VENC was appropriate for the physical flow phantom because the peak velocities were known to be <10 cm s$^{-1}$. The physical flow phantom was also imaged using a standard multi-planar PC-SPGR acquisition for comparison with a reference MRI measurement and validation of the proposed method. Three coplanar stacks with orthogonal velocity-encoding directions were acquired using the PC-SPGR acquisition, with voxel resolution and FOV identical to the bSSFP stacks. The PC-SPGR stacks were acquired with a matched VENC$_{spgr}$ = 30 cm s$^{-1}$. Both MRI methods were compared against a measurement of volumetric flow rate using a measuring cylinder, which was treated as the gold-standard.

For comparison of the two acquisitions, all magnitude stacks for both sequences were rigid registered (MIRTK, BioMedIA, UK) with cubic spline interpolation to one of the PC-SPGR stacks. The registration parameters were then applied to the equivalent phase data to align all stacks. bSSFP velocity volumes were reconstructed using the proposed method. PC-SPGR velocity volumes were reconstructed by standard vector addition of phase data. The flowing water in the plastic tube was segmented by thresholding the SPGR magnitude data. Voxel-wise comparison of measured velocity values between the two acquisition methods was performed and the mean bias was calculated across all voxels within the plastic tube. Cross-sectional ROIs were manually drawn at 20 locations throughout the phantom and used to calculate the flow rate measured with both MRI methods, which were compared against the measuring cylinder measurement.

**In utero fetal study**. All human fetal imaging protocols were performed in compliance with ethical regulations (REC 14/LO/1806) and with approval from the local NHS London Bridge Research Ethics Committee. All participants gave written informed consent prior to enrollment. Seven singleton pregnancies were scanned. The only selection criteria was the availability of five non-coplanar bSSFP stacks for reconstruction. This was to ensure extensive sampling of all possible directions of blood flow in the heart and to allow for redundancy needed for stable inversion, even in the event of fetal and/or maternal motion which introduces unpredictable rotations to the actual velocity encoding directions. Typically, 7–11 slices were acquired per stack depending on the size of the fetus and the orientation of the stack. Full details of the fetal subjects, who ranged from 24 to 33 weeks GA, are given in Supplementary Table 1.

Sequence parameters had been optimized as described in van Amerom et al.[33] to balance good signal with necessary spatio-temporal resolution, while ensuring full coverage of fetal and maternal anatomy within the field of view and adhering to safety constraints. The safety constraints were whole body SAR < 2.0 W/kg[55], low PNS (equivalent to d$B$/d$t$ < 60% of mean PNS perception threshold[56]) and sound pressure level < 85 dB(A) experienced by the fetus, accounting for >30 dB attenuation in utero[57]. All data were acquired on a 1.5 T Ingenia MRI scanner (Philips, Netherlands) using an anterior torso coil array in combination with a posterior spine coil array to measure signal in 28 receiver channels.

The bSSFP sequence was run with regular Cartesian $k$–$t$ undersampling[35] with: TR/TE 3.8/1.9 ms, flip angle 60°, FOV $400 \times 304$ mm, voxel size $2.0 \times 2.0 \times 6.0$ mm, 8× acceleration, 72 ms temporal resolution, 96 images per slice, slice overlap 2–3 mm, VENC$_{bssfp}$ = 159 cm s$^{-1}$. Coil calibration data were acquired prior to bSSFP acquisition and $k$–$t$ training data were acquired following the under-sampled acquisition. Acquisition time of a single stack was typically 155 s.

$k$–$t$ SENSE reconstruction of bSSFP data was performed in MATLAB (Mathworks, USA), with additional functionality from ReconFrame 3.0.535 (GyroTools, Switzerland). The 4D flow cine reconstruction framework was implemented using a combination of MATLAB and both the Image Registration Toolkit (IRTK, v1.0, BioMedIA, UK) and the Medical Image Registration Toolkit (MIRTK, v2.0.0, BioMedIA, UK), extending work by Kuklisova et al.[28], van Amerom et al.[32,33], and Deprez et al.[52]. In keeping with the 4D magnitude cine framework, 4D flow cine volumes were reconstructed with an isotropic spatial resolution of 1.25 mm and $N_h = 25$ cardiac phases. The code underlying the proposed framework can be found online at https://github.com/mriphysics/fetal_cmr_4d.

**Evaluation of 4D flow cine volumes**. Whole heart 4D flow cine volumes for each subject were assessed independently by two expert MRI fetal cardiologists (Reader 1 = MvP, Reader 2 = DL with 3- and 5-years of experience reading fetal cardiac MRI, respectively). MRtrix3[58] (v3.0) and Paraview[59] (v5.4.1) were used to visualize the hearts in any orientation with optional overlay of velocity vectors on the anatomical cine images. For the purpose of calculating temporal blood flow curves, single cross-sectional 2D ROIs were manually drawn perpendicular to selected blood vessels using the Medical Imaging Interaction Toolkit (MITK) Workbench software (v2016.11.0, German Cancer Research Center, Germany) which allowed free-rotation of the 4D magnitude cine volume reconstruction. The following vessels were sampled: ascending aorta (AAo), descending aorta (DAo), pulmonary artery (PA), superior vena cava (SVC), and ductus arteriosus (DA). Voxel-wise blood flow was calculated as the product of velocity vector magnitude ($|\mathbf{V}|$) and voxel area ($1.25 \times 1.25$ mm). Vessel blood flow was calculated as the sum of all voxels within the ROI. These measurements were normalized to fetal weight, which was calculated based on expert manual segmentation of the fetus[60] using structural scans acquired in the same session.

The seven volumetric 4D datasets were anonymised and presented to each expert reader for analysis of mean flow values in each major vessel (AAo, DAo, PA, SVC, and DA), as well as for assessment of image quality and arterial flow curve reliability. The readers reoriented the 4D magnitude volumes to provide perpendicular cross-sections for each vessel and then drew a contour delineating the vessel perimeter, as close as possible to an agreed anatomical landmark to maximize consistency. The agreed ROI landmark locations were: AAo = at the level of the right pulmonary artery, DAo = behind the heart at the level of left atrium, PA = proximal to the bifurcation of the branched pulmonary arteries, SVC = at the level of the three-vessel view, DA = mid-vessel adjacent to the trachea.

This task was repeated (i.e., two trials) with the subject order randomized for each trial. One ROI was drawn per vessel at the same timepoint, which was applied to all cardiac frames. The ROIs were used to extract flow rates from the velocity magnitude volumes—this derived data was further assessed without reference to the flow maps themselves. The expert readers scored the data in two ways:

1. Confidence for accurate delineation of each vessel was rated using a 1–3 scale where: 3 = entire vessel boundary clearly defined, 2 = partial vessel boundary clearly defined, 1 = vessel boundary poorly visualized. If an ROI could not be drawn, the vessel was not scored. The ROI failure rate (%) was defined as the total number of ROIs that were not drawn divided by the total number of possible vessel segmentations ($n = 140$).
2. Phasic variation of all arterial flow curves using a 1–3 scale where: 3 = clear systolic and diastolic phases, 2 = clear systolic and diastolic phases with some variation in the flow curve, 1 = limited phasic variation with unclear systolic and diastolic phases. For arterial flow curves with no phasic variation, such as if the curve was flat throughout the cardiac cycle, the vessel was not scored. The flow curve failure rate (%) was defined as the total number of arterial flow curves which displayed no phasic variation divided by the total number of arterial vessel ROIs. Note, the SVC is a venous vessel which exhibits minimal phasic variation, hence it was excluded from this part of the analysis.

**Reporting summary**. Further information on research design is available in the Nature Research Reporting Summary linked to this article.

## Data availability
The data that support the findings of this study are available upon reasonable request from the corresponding author (T.A.R.). Patient data are not publicly available due to them containing information that could compromise research participant privacy or consent. An anonymised version of the data pertaining to subject ID03 is available from https://doi.org/10.6084/m9.figshare.c.4689437. Source data are provided with this paper.

## Code availability
The simulated flow phantom code is available at https://github.com/tomaroberts/synthflow_phantom. The 4D cine reconstruction code is available at https://github.com/mriphysics/fetal_cmr_4d.

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

## Acknowledgements
Thank you to Joanna Allsop, Elaine Green and Ana Gomes for scanning of volunteers and patients. We would also like to thank the funding bodies which have supported this project. This work was supported by the Wellcome/EPSRC Centre for Medical Engineering [203148/Z/16/Z], Wellcome Trust IEH Award [102431]; the Engineering and Physical Sciences Research Council [EP/H046410/1]; and the Medical Research Council Strategic Fund [MR/K0006355/1]. The research was funded by the National Institute for Health Research (NIHR) Biomedical Research Centre based at Guy's and St Thomas' NHS Foundation Trust and King's College London and supported by the NIHR Clinical Research Facility (CRF) at Guy's and St Thomas'. The views expressed are those of the author(s) and not necessarily those of the NHS, the NIHR or the Department of Health.

## Author contributions
T.A.R. and J.F.P.vA. were responsible for study design, data collection, methods development, software development, data analysis, and paper preparation. A.U. performed methods and software developments. D.F.A.L. and M.P.M.vP. performed expert image analysis and provided the clinical interpretation. A.P. performed sequence and software developments. J.-D.T. performed software developments. C.A.M. performed additional data reconstruction and the image analysis. L.J.H. performed software developments and contributed to the paper preparation. S.J.M. contributed to methods development. K.P., M.A.R., and R.R. enabled the patient recruitment and provided clinical oversight. M.D. provided image reconstruction expertize and methods development. J.V.H. was responsible for the study design, methods development, data interpretation, and paper preparation.

## Competing interests
The authors declare no competing interests.
