## [Peer Review File · Nature Communications]

Reviewers' comments:

Reviewer #1 (Remarks to the Author):

This manuscript describes the implementation and initial validation of a new MRI approach to conduct large volume flow sensitive MRI studies in the fetus that can overcome significant fetal motion during such lengthy scans. The framework of the novel approach is introduced and proof of principle studies are presented that include a numerical simulation, in vitro experiments with a tube and constant flow and in 7 human subjects.

The strengths of the work are the high level of innovation, the potential high impact for research and clinical use of 4D Flow MRI in the fetus, the attempt to demonstrate feasibility in silico, in vitro, and in vivo, and good quality visualizations of the velocity vector fields. Innovations such as described here are likely necessary to transition the powerful 4D Flow MRI to the unique challenges of fetal imaging. The way that bSSFP imaging is used here to provide anatomical and quantitative flow information is clever and the motion correction scheme is well executed. It is also noted with enthusiasm by this reviewer that the code developed here has been made public in github repositories.

However, weaknesses are also noted in the form of lack of rigor in the analysis methods, which reduce the enthusiasm for the manuscript in its current form.

Major concerns

1) The authors do not make a convincing case of why they chose to pursue bSSFP 4D flow MRI instead of the much more common SPGR 4D Flow MRI. I don't see any reason why the same proposed framework would not have worked with SPGR. bSSFP has certain advantages (higher SNR/contrast, shorter TR), but also disadvantages: sensitivity to off-resonance, higher SAR, etc). A more nuanced discussion about the pros and cons seems appropriate. As written, it seems that bSSFP was chosen simply because the team had successfully used it previously for cardiac function imaging.

2) Page 11: Background phase corrections: A more thorough explanation for the source of well studied background phase errors seems appropriate (eddy currents, gradient timing errors, etc), especially since it seems more dominant than in standard PC MRI (Fig 3c and d). Also, it is not intuitive to me why the amniotic fluid can be used as 'static background tissue' for background phase estimation. The fluid is likely moving (maternal respiratory motion and any fetal motion), which makes the phase of the fluid instable.

3) Page 15: Velocity Drift Correction: It is not clear why the drift occurs and why it is not eliminated by the background phase correction. It appears that the background phase correction should eliminate this offset.

4) Other corrections for quantitative phase contrast measures that have been found to be essential are neither included nor discussed, e.g. Maxwell term corrections (e.g. Lorenz 2014, PMID: 24006013).

5) A more detailed discussion of the limitations of the spatial and temporal resolution is warranted: the acquired resolution is 2x2x6 mm, the interpolated to 1.25x1.25x1.25 and the temporal resolution is 72 ms and then interpolated to 25 cardiac phases. What are the dimension of the imaged structures (simulated and physical phantom, expected sizes of fetal vessels). It appears that even the interpolated voxel size might only cover a very few voxels of an in vivo vessel, much less so the acquired voxel size. I believe that the visualizations of the vector plots (e.g. Figs 6 and 7) are even further interpolated. I also believe I can see blurring of the vessel boundaries in Fig 4 c. The heart rate of the fetuses is not provided, but I would suspect it is rather high and a temporal resolution of 72 ms is probably corresponding to less than 10 frames per cardiac cycle.

6) Page 19: Why is the IVC not analyzed?

7) Page 22/23: Analysis of the physical flow phantom: I don't find the metrics for the validation very effective. It is limited to average bias for the velocity directions for all voxels in vessels. If this measure would be very noisy but with a mean of 0, it would still look excellent with this metric. A more meaningful metric would be a mean square error. An equally meaningful measure would have been the analysis of the flow rate and velocity distribution at various tube locations against the flow rate as measured with a gold standard (ultrasound sensor, bucket, ...).

8) Page 27: 'however, the relative absolute flow rates of the vessels were consistent ...' This statement needs more explanations. It appears that the measures are not that consistent: the std in the normalized blood flows are rather big (Fig 8b, e.g. see DA)).

9) Kt-accelerated 4D flow MRI Scans have been thoroughly investigated. They typically underestimate peak flow. Some references / discussion to this work seems adequate.

10) The in vivo analysis is very limited. I would have liked to see some additional information beyond simply the flow measures, which had large variances. For example, the expert readers could have scored image quality and confidence in proper delineation of the target vessel areas. Was any velocity aliasing observed? How about repeatability of the analysis (inter and intra observer). How about repeatability of the measures (scan twice)?

11) There is unconcise and unconventional terminology used throughout the manuscript:

a) Page 5: three-dimensional phase contrast MRI – should be three-directional, see also page 7: '3D velocity mapping' and '3D flow vectors' and 'three velocity dimensions'– that should all refer to a 3-directional vector and more.

b) Page 8: multiple dynamics (frames) obtained for each slice: dynamics seems an unusual term to describe acquisitions throughout multiple cardiac cycles – 'frames' or 'time frames' seems more appropriate.

c) Figure 2 legend: real-time accelerated bSSFP: the term 'real-time' is misleading. I believe what the authors mean is 'beat-to-beat' accelerated. The images are not displayed in realtime as they would

eb e.g. in X-ray or ultrasound realtime imaging. Even if a hardware platform would allow for realtime MRI, a kt-accelerated acquisition uses data from future time frames and , hence, realtime display would not be possible.

d) The term 'velocity volume' and 'velocity component volume' and 'velocity magnitude volume' is repeatedly used for velocity measures, not flow measures. This seems to be an unconventional term that I can not find defined anywhere.

Minor issues

12) The authors don't define what 4D flow MRI is until page 4 last paragraph (after abstract and 1.5 pages of intro). Yet other terms such as 4D cine imaging of the fetal heart are used. I suggest to define 4D flow MRI much earlier and in the abstract since the term '4D Flow MRI' is somewhat ambiguous. Here, a dynamic, volumetric acquisition of the 3-directional velocity vector field is captured.

13) Ref 15: source unknown

14) Page 3, paragraph2: '... however, these approaches have been limited to single-slice ...' – that does not seem to be true for ref 15 (see comments above for incomplete ref 15)

15) Ref 16: incomplete (last author is missing)

16) Together with ref 16: MacDonald et al demonstrated 4D flow in the fetus of the rhesus macaque at the same time as ref 16 (PMID: 30102431)

17) Page 4/ page 5 and/or Page 7: It should be noted that a very clever 3-directional bSFFP PC approach had been previously introduced : Santini 2009 (PMID: 19585606)

18) Page 6, Fig 1: It strikes me as odd that the figures here do not use a standard right hand coordinate system. More importantly, it seems that the labels for the axes should not be x, y , and z but a moment space (or spatial and moment space?)

19) Page 9: grammar in Figure 2: Background phase is estimated and subtracting ...

20) Page 16: Simulated Flow Phantom: what Venc was used in the simulations?

21) Page 18: How many slices are in one stack?

22) Page 23: 'normal spiral relationship': find that hard to see Maybe some arrows would help?. I also believe the authors mean 'helical' flow which is a more common description of flow in the ascending aorta.

Reviewer #2 (Remarks to the Author):

Dear Madams and Sirs,

thank you for giving me the opportunity to review this interesting article entitled:

“Fetal Whole-Heart Blood Flow Imaging Using 4D cine MRI”.

It is an indeed a challenge to use the magnetic resonance tomography (MRI) for cardiovascular imaging (cMRI) in the fetus in utero, especially for blood flow imaging using the MRI. The authors are aware of the main problem of cMRI, the triggering of the fetal heart beat, which is necessary for both cardiac imaging and for flow measurement within the fetus. Recent studies have shown that Doppler Ultrasound Triggered flow imaging for 2D and 3D cine MRI (e.g. Kording et al. 2018, Schoennagel et al. 2019) were applicable in human fetus, which the authors have not taken in account. Furthermore, at the annual meeting of the ISMRM 2019 as well as at the RSNA 2019, the same group has shown the feasibility of 4D cine MRI, which abstracts are also available online. The authors use the so-called “slice-to-volume registration” for solution and seem to succeed in flow measurement. This method is a postprocessing method and is comparable with the metric optimized gating technique, which the authors should mention in their text.

The main weakness of this whole manuscript is the lack of gold standard. Usually the ultra-sound examinations are used in the clinical routine in obstetrics and thus should be used to compare the flow velocity within the different vessels mentioned in this manuscript. The use of 4D flow measurement is very promising, bur from clinical point of view this post processing, rather complicated method is not really a solution for the daily use. It is also not very clear, which advances can be obtained with this rather mathematical method compared to ultrasound, which is readily available in most OB-Gyn departments. It seems that there are other prospective triggering methods that are probably easier to use for the consumer. To be published, the authors should include more patients. The authors should also cite some papers which have been dealing with the triggering method, especially, because there are some approach for fetal cardiac MRI using an MR-compatible cardio-tocograph (CTG) as an prospective, external triggering method. The statistical analyses are not very exact and should be more precise, especially the inter-rater reliability.

Overall, the idea behind this manuscript is highly interesting, but it needs to be worked over again, especially the clinical impact is not clear.

Dear Reviewers,

Please find below point-by-point responses to your comments. Thank you for taking the time to comment on our manuscript – we think your contributions have greatly strengthened the paper. We are aware that some of our responses are quite lengthy – we wanted to be thorough and provide as much information as possible.

Changes made to the manuscript in response to the comments can be found in two locations. The manuscript document contains inline comments denoting the reviewer and the number of the corresponding comment (e.g.: Reviewer 2, comment 4 = [R2.4]). In this document, manuscript changes are referenced by page number and the number corresponding to the first line of the alteration (e.g.: p 12, line 16).

In summary, the major improvements to the manuscript are, but not limited to:

- 1. New 4D flow visualisations in an older, larger fetal heart with right aortic arch. The original manuscript only included images from one fetus: we hope this new dataset provides further evidence that our method is successful and robust. The new visualisations also clearly illustrate the anatomical and hemodynamic changes associated with fetal development: the heart is larger and the blood flow velocity vectors are considerably faster.**
- 2. New expert reader scoring of ROI delineations, repeatability analysis of *in utero* 4D flow quantification, and assessment of arterial flow curve phasic behaviour.**
- 3. New validation of the physical flow phantom against gold-standard benchtop flow measurement.**
- 4. Improved methods section including greater clarification on sources of phase change within the acquired images**
- 5. Overhauled Introduction and Discussion sections with greater emphasis on comparisons with Doppler ultrasound flow measurements and existing MRI flow methods.**

Once again, thank you for taking the time to review our paper.

**Kind regards,
Tom Roberts**

Reviewer #1:

This manuscript describes the implementation and initial validation of a new MRI approach to conduct large volume flow sensitive MRI studies in the fetus that can overcome significant fetal motion during such lengthy scans. The framework of the novel approach is introduced and proof of principle studies are presented that include a numerical simulation, in vitro experiments with a tube and constant flow and in 7 human subjects.

The strengths of the work are the high level of innovation, the potential high impact for research and clinical use of 4D Flow MRI in the fetus, the attempt to demonstrate feasibility in silico, in vitro, and in vivo, and good quality visualizations of the velocity vector fields. Innovations such as described here are likely necessary to transition the powerful 4D Flow MRI to the unique challenges of fetal imaging. The way that bSSFP imaging is used here to provide anatomical and quantitative flow information is clever and the motion correction scheme is well executed. It is also noted with enthusiasm by this reviewer that the code developed here has been made public in github repositories.

However, weaknesses are also noted in the form of lack of rigor in the analysis methods, which reduce the enthusiasm for the manuscript in its current form.

Major concerns

1) The authors do not make a convincing case of why they chose to pursue bSSFP 4D flow MRI instead of the much more common SPGR 4D Flow MRI. I don't see any reason why the same proposed framework would not have worked with SPGR. bSSFP has certain advantages (higher SNR/contrast, shorter TR), but also disadvantages: sensitivity to off-resonance, higher SAR, etc). A more nuanced discussion about the pros and cons seems appropriate. As written, it seems that bSSFP was chosen simply because the team had successfully used it previously for cardiac function imaging.

[R1.1] The reviewer is correct in suggesting that the proposed framework would work with SPGR. For fetal imaging, SNR is a major problem because the fetus is distant from the RF coils and the field-of-view must be large enough to avoid phase wrap-around from the mother. Hence, the higher SNR of a bSSFP acquisition compared to an SPGR acquisition is highly desirable. An SPGR-based acquisition may produce phase maps of sufficient quality to reconstruct 4D flow volumes, but the anatomical images would be compromised. The beauty of the bSSFP-based acquisition is that it provides both capabilities: high quality 4D anatomical imaging and high quality 4D flow data.

Additional technical considerations, such as those outlined by the reviewer, must be taken into account when using bSSFP, but they are very manageable at 1.5T, which is why we deliberately chose to operate at this field strength. Moving to 3T would require greater consideration of these factors.

There is also a difference in background phase behaviour between SPGR and bSSFP, with the latter being intrinsically robust against static field variation (except close to black bands, which we can easily avoid in the fetal imaging case at least at 1.5T).

There are also some potential challenges for SPGR in this application. It has already been noted that the SNR is lower in SPGR. However, in SPGR quantitative flow (q-

flow) applications it is common to exploit inflow enhancement to boost SNR in the vessels where accurate phase measurement is required. In the current approach, we expect to acquire slices in diverse, but arbitrary planes relative to the local flow direction. Thus the degree of inflow enhancement is likely to be different between different geometries and this could have two adverse effects: 1) the precision of the phase measurement may be lower in some geometries than in others, compromising overall performance; and 2) the signal magnitudes for the same blood volume might be very variable between geometries, which could impair the vital registration step in our approach. We would remark that we have not tested either of these conjectures *in utero* as we have only acquired very limited SPGR images in fetuses in this study.

We took the view that it was essential to validate the method as used (with its reliance on intrinsic flow sensitivity combined with a diversity of imaging planes) against classic SPGR quantitative flow imaging using additional gradients, but after that either sequence could be deployed. In actual fetal applications, there is pressure on examination time so, as noted above, deploying a strategy that delivers classic anatomical cine imaging equivalent to any cardiac examination *in utero* or *ex utero* as well as providing quantitative flow data offers strong advantages. There would need to be a strong motivation for potentially doubling the examination time used for cine imaging if it is feasible to get high quality data for both structural and flow assessment from one acquisition.

We have updated the manuscript to include more discussion of why bSSFP was used and described some technical considerations when using the sequence [p 5, line 8 // p 6, line 3].

2) Page 11: Background phase corrections: A more thorough explanation for the source of well-studied background phase errors seems appropriate (eddy currents, gradient timing errors, etc), especially since it seems more dominant than in standard PC MRI (Fig 3c and d). Also, it is not intuitive to me why the amniotic fluid can be used as ‘static background tissue’ for background phase estimation. The fluid is likely moving (maternal respiratory motion and any fetal motion), which makes the phase of the fluid instable.

[R1.2] Thank you for this and other comments related to sources of phase offset. These comments inspired us to re-examine our data and to more thoroughly investigate the source of the phase roll in Figure 3.

Following the reviewer’s comments (point R1.4 below) we investigated the effect of concomitant gradients on our phase data, by following the theory outlined by Bernstein et al. (1998) [PMID: 9469714]. In that paper, the authors demonstrate the z^2 phase error caused by a G_y gradient (Figure 4, in the paper). With their sequence and hardware, the phase error given by concomitant gradient coefficient $A = 0.499$ degrees/cm. On our scanner, the fetal heart is confined to a $\pm 10\text{cm}^3$ region centred on the isocenter because of the maternal habitus. For Bernstein et al, a distance of 10cm from isocenter results in a phase error of $0.499 * 10^2 = 49.9$ degrees.

We calculated the equivalent phase error for the gradient parameters of our *in utero* imaging sequence, yielding: $A = 0.07$ degrees/cm. At 10cm from isocenter, the phase error = 7 degrees. Thus although there could be a detectable contribution to the

background phase coming from this effect, it is clearly not the dominant contribution in our data. Concomitant gradients are inherently small in fetal imaging because we use relatively benign gradient configurations compared to unconstrained adult imaging to minimise acoustic noise and ensure there is no peripheral nerve stimulation in the mother.

With further investigation we established that a dominant source of the phase roll in Figure 3 in the paper was due to the body coil SENSE reference scan used as part of the k-t SENSE. In our reconstruction, which directly mirrors the behaviour of the original methods proposed by Pruessmann [PMID: 10542355] and Tsao [PMID: 14587014], the final images are modulated by the amplitude and phase distributions of the reference body coil image that is used to divide out the anatomy to determine the relative sensitivities of the elements of the receiver array. The SENSE reference scan uses a spoiled gradient echo (SPGR) sequence, which has a markedly different phase evolution to a bSSFP acquisition. The phase difference between these two sequences was effectively imprinted into the k-t SENSE reconstructed images. For example, in the figure below, panel b) shows an uncorrected bSSFP acquisition phase image after k-t SENSE reconstruction, panel c) shows the SPGR body coil phase image and panel d) shows the subtraction, which now has much more uniform phase across the image.

Reviewers Figure R1: (a) Magnitude image, (b) Uncorrected bSSFP phase image, (c) SPGR SENSE reference phase image, (d) Subtraction: panel b – panel c.

We have explored removing the phase of the body coil reference image from the reconstruction, but found that there are residual phase variations in some locations in some images (see Reviewers Figure R2a), so this approach does not provide optimal correction in all cases. We concluded that, as noted by other authors [Nielsen, PMID: 19230016], there are a multitude of contributions to the background phase and so our pragmatic approach of using a data driven correction currently remains the most robust strategy we have found (Reviewers Figure R2b).

Reviewers Figure R2: Phase corrected images using (a) Subtraction of SPGR reference scan only. Note, this is equivalent to Figure R1d but it has been windowed for the benefit of this comparison. Red arrows indicate residual phase variations. (b) Polynomial correction, as in manuscript.

In light of this comment, we believe Figure 3 was misleading in its original form: the narrow windowing in Figure 3c was intended to accentuate the phase roll, but unintentionally made the phase roll look very severe. Instead, we now include a phase image showing the entire uterus and we have adjusted the windowing on the zoomed in panels [p 14, line 1].

We also agree that the phrase “static background tissue” is misleading terminology. The reviewer is correct in saying that the amniotic fluid is non-static, however, the phase changes in this region are low and on a large scale, whereas the phase changes in the heart are greater and highly localised. The polynomial phase correction removes the large scale changes whilst preserving the localised changes. This can be seen in Figure 3f, where the phase in the vessels is clearly different from the background, which is highly uniform and close to zero. We have made this clearer in the manuscript [p 13, line 12].

3) Page 15: Velocity Drift Correction: It is not clear why the drift occurs and why it is not eliminated by the background phase correction. It appears that the background phase correction should eliminate this offset.

[R1.3] The dominant causes of the phase changes in the data are due to fetal and maternal motion. As mentioned above, the background phase correction deals with the large scale effects, but if the fetus moves there is an additional motion-induced phase change on top of the flow velocity. This is the velocity we are trying to correct with the

drift correction. We do not have a fixed stance on this – this was an empirical observation which we thought was reasonable to try to correct. We have updated the Methods to clarify this [p 17, line 12] and included additional comment in the Discussion [p 38, line 7].

4) Other corrections for quantitative phase contrast measures that have been found to be essential are neither included nor discussed, e.g. Maxwell term corrections (e.g. Lorenz 2014, PMID: 24006013).

[R1.4] This comment ties in with the previous comments regarding sources of phase change in the data. As already noted above, for fetal imaging we do not drive the gradients hard, which means that the effects of concomitant gradients, eddy currents and gradient non-linearity are more benign.

The effect of concomitant gradients on our system has already been described above in response to comment 1.2.

Eddy currents can be a problem for bSSFP sequences if the imaging gradients are driven too aggressively. As Tsao et al. [PMID: 15906282] showed, pushing the gradients too hard leads to overt artefacts due to the eddy currents (Figure 7b). For their experiments, Tsao et al. used shorter TE/TR times (1.41/2.8 ms) and a smaller FOV (320 x 272 mm) compared to our *in utero* scans (TE/TR = 1.9/3.8 ms, FOV = 400 x 304 mm) to demonstrate the effects of eddy currents. We have been able to replicate eddy currents effect in other scans, but for our fetal bSSFP sequence we have found that if we extend TR, the sequence is stabilised.

Our scanner includes a geometric correction option for addressing gradient non-linearity, which is already implemented in our reconstruction pipeline.

We have updated the Theory [p 13, line 2 // p 13, line 12] and Discussion [p 36, line 18] sections of the manuscript to discuss these effects.

5) A more detailed discussion of the limitations of the spatial and temporal resolution is warranted: the acquired resolution is 2x2x6 mm, the interpolated to 1.25x1.25x1.25 and the temporal resolution is 72 ms and then interpolated to 25 cardiac phases. What are the dimension of the imaged structures (simulated and physical phantom, expected sizes of fetal vessels). It appears that even the interpolated voxel size might only cover a very few voxels of an *in vivo* vessel, much less so the acquired voxel size. I believe that the visualizations of the vector plots (e.g. Figs 6 and 7) are even further interpolated. I also believe I can see blurring of the vessel boundaries in Fig 4 c. The heart rate of the fetuses is not provided, but I would suspect it is rather high and a temporal resolution of 72 ms is probably corresponding to less than 10 frames per cardiac cycle.

[R1.5] The reviewer is correct on these points. We had tried to cover them in the original Discussion, but given the reviewer's comments, we clearly had not done so sufficiently. We have updated the manuscript to include a more thorough discussion of these limitations [p 38, line 10].

6) Page 19: Why is the IVC not analyzed?

[R1.6] It is not normal practice in fetal medicine to analyse flow through the IVC (for example, see Prsa et al. 2014 [PMID: 24874055]) because the local geometry is complicated compared to adult hearts, so we didn't include the IVC as we didn't think readers would find this informative. The conventional reason for this is that the hepatic vessels drain directly adjacent to the IVC as it enters the right atrium and hence there is no reliable measurement plane that would include all venous return to the fetal heart. Instead, the descending aortic flow includes all flow to the lower body of the fetus and total placental blood flow. This value is therefore equivalent to total inferior venous return as would be measured in adults.

7) Page 22/23: Analysis of the physical flow phantom: I don't find the metrics for the validation very effective. It is limited to average bias for the velocity directions for all voxels in vessels. If this measure would be very noisy but with a mean of 0, it would still look excellent with this metric. A more meaningful metric would be a mean square error. An equally meaningful measure would have been the analysis of the flow rate and velocity distribution at various tube locations against the flow rate as measured with a gold standard (ultrasound sensor, bucket, ...).

[R1.7] We thank the reviewer for highlighting the shortcomings of this analysis. We have updated the analysis to include normalised root mean square error [p 25, line 12]. We also now include flow measurements calculated using both MRI methods and compare to a gold standard, namely a volumetric flow determined using a measuring cylinder.

We consulted with some of our ultrasound colleagues about flow quantitation in the phantom using ultrasound and they said it would be extremely challenging and perhaps not possible because the materials (plastic tubing, glass) were selected for MRI compatibility, but are not compatible with ultrasound. The spherical glass bottle would cause multiple reflections and wave mode conversions which would attenuate the signal. In principle, it might be possible to image the flow within the tubing if it was removed from the spherical glass flask and located outside of the scanner, but success is not guaranteed as it depends on the plastic used in the tubing. Furthermore, it would not be possible to perform MRI with this configuration as the surrounding non-flowing water and its spherical outer surface geometry is needed to minimise susceptibility artefacts. A paired measurement would not be possible and performing an entirely new ultrasound experiment brings its own set of challenges and limitations, with attendant problems in achieving a reliable and robust quantitative comparison.

Instead, we now include a measurement of volumetric flow using a measuring cylinder. We compare this against MRI-derived flow values measured using both the reference standard PC-SPGR method and our bSSFP method. Both MRI methods showed good agreement with the cylinder measurement. We have updated the methods [p 19, lines 11 and 21], Figure 5 [p 26] and the corresponding section of the Results [p 25, line 15].

8) Page 27: 'however, the relative absolute flow rates of the vessels were consistent ...' This

statement needs more explanations. It appears that the measures are not that consistent: the std in the normalized blood flows are rather big (Fig 8b, e.g. see DA)).

[R1.8] We have revised this statement to say: "... the left and right outflow tracts demonstrated fast, pulsatile flow whilst the superior vena cava demonstrated slower, more uniform flow consistent with venous return." [p 34, line 15]

9) Kt-accelerated 4D flow MRI Scans have been thoroughly investigated. They typically underestimate peak flow. Some references / discussion to this work seems adequate.

[R1.9] Thank you for these references – they provide evidence to partially explain why our flow measurements underestimate literature values. We have now commented on this in the Discussion, including these references. [p 38, line 11]

10) The in vivo analysis is very limited. I would have liked to see some additional information beyond simply the flow measures, which had large variances. For example, the expert readers could have scored image quality and confidence in proper delineation of the target vessel areas. Was any velocity aliasing observed? How about repeatability of the analysis (inter and intra observer). How about repeatability of the measures (scan twice)?

[R1.10] Thank you for this comment, which was similar to a comment made by Reviewer #2. This was clearly a deficiency in the first manuscript, so we have done some additional analysis based on two independent expert readers, as outlined below. However, we cannot include scan-rescan measurements as these were not feasible for this study because the scanning protocol was already at capacity. Also image quality assessment of anatomical features was performed in our previous paper [PMID: 31081250] so is not repeated here.

Here, we now include expert reader vessel delineation scores, intra- and inter-expert reader repeatability analysis and assessment of arterial flow curves. There were three parts to this analysis:

- 1) Two expert readers were blinded to the data and asked to draw ROIs in each vessel in a predefined list for each subject using the magnitude cine volumes. The readers were each presented with the data twice in random subject order for intra-reader repeatability analysis. Expert readers were asked to rate their confidence for accurate delineation of each vessel on a scale of 1-3 [3 = entire vessel circumference clearly defined, 2 = partial vessel circumference, 1 = vessel poorly visualised]. If an ROI could not be drawn, the vessel was scored 0 representing a fail.**
- 2) The ROIs were transferred to the velocity maps and used to calculate mean flow values in each vessel. Intra- and inter-reader repeatability analysis was performed to assess measurement variation due to ROI drawing and reader bias.**
- 3) The vessel flow curves generated from the ROIs in arteries were presented to the expert readers, again in a blinded fashion (random order and with no subject identifiers). The expert readers were asked to assess the quality of the arterial flow curves by rating the phasic variation on a scale 1-3 [3 = clear systolic and diastolic phases, 2 = clear phases with some variation in flow curve, 1 = some phasic variation, but unclear systolic and diastolic phase]. A score of 0**

represented a failed flow curve, for example if the curve was flat throughout the cardiac cycle when it was expected to be pulsatile. The SVC was excluded because it is a venous return vessel hence phasic variation is difficult to assess.

For part 1), there was a 3% failure rate when drawing the ROIs. Both expert readers, in both trials, could not confidently draw the ROI for the ductus arteriosus in the smallest fetus (24⁺² weeks). The expert readers showed good agreement when drawing ROIs for all other vessels. The ductus arteriosus was the most difficult vessel to visualise, with the lowest mean score of 1.1. All other vessels were rated 2 or higher by both readers.

In part 2), Bland-Altman analysis of mean flow values showed small intra-reader biases of 6% and 2%, with bias standard deviations of 23% and 21%, respectively. As expected, the inter-reader bias was slightly larger at -13% with a standard deviation of 28%. In summary, the self-biases were small implying the expert readers were consistent with their ROI drawing, whereas the standard deviations were larger, suggesting that the variation is mainly due to limitations of spatial and temporal resolution. This is consistent with Figure 9c, as shown previously. Increasing the spatial and temporal resolution of our method is a focus for future work, which we talk about in the Discussion.

For part 3), 4% of the arterial flow curves showed no phasic behaviour. There are various potential causes in these cases including: incompletely corrected fetal motion, partial volume effects, regions not sampled by all stacks and ROI placement.

Overall, we were encouraged by these results and we thank the reviewers for prompting this analysis. The 4D magnitude volumes allow the expert readers to quickly draw ROIs in multiple vessels, which they are able to do with excellent repeatability. The vast majority of vessels segmented demonstrated phasic flow behaviour. The ductus arteriosus is a challenging vessel to measure, which we anticipated, and this is reflected in the lower flow curve ratings. Given that the underlying q-flow method is well established and we validate our approach against a standard implementation, it is most likely that the broader variation in the data is due to limitations of spatial and temporal resolution, which we are actively working to improve.

This analysis is now described in the Methods [p 22, line 1 onwards], the corresponding section of the Results has been re-written [p 31-32, line 14 onwards], Figure 9 has been updated, [p 33], Table 2 has been added [p 49] and the Discussion has been updated [p 36, line 10 // p37, line 22].

11) There is unconcise and unconventional terminology used throughout the manuscript:
a) Page 5: three-dimensional phase contrast MRI – should be three-directional, see also page 7: ‘3D velocity mapping’ and ‘3D flow vectors’ and ‘three velocity dimensions’ – that should all refer to a 3-directional vector and more.

[R1.11a] Thank you for pointing this out. We have updated these terminology throughout the manuscript.

b) Page 8: multiple dynamics (frames) obtained for each slice: dynamics seems an unusual term to describe acquisitions throughout multiple cardiac cycles – ‘frames’ or ‘time frames’ seems more appropriate.

[R1.11b] We have updated this in the text.

c) Figure 2 legend: real-time accelerated bSSFP: the term ‘real-time’ is misleading. I believe what the authors mean is ‘beat-to-beat’ accelerated. The images are not displayed in realtime as they would be e.g. in X-ray or ultrasound realtime imaging. Even if a hardware platform would allow for realtime MRI, a kt-accelerated acquisition uses data from future time frames and , hence, realtime display would not be possible.

[R1.11c] We do accept the referees point that real time display of these images during live examination would not be feasible for the reasons they state. However, we do believe that ‘real-time’ is fair terminology for our acquisition because we get a rapid series of sequential images that asynchronously depict the beating fetal heart. Changes in fetal heart rate and other ‘real time’ events such as maternal breathing or changes in maternal and/or fetal pose are faithfully reproduced in the sequence that they occurred and the same relative timings. For the fetal heart, the k-t SENSE method achieves a temporal resolution close to the acquisition time per undersampled k-space update (72 msec in our data) because the reconstruction has lower time resolution in other areas (away from the fetal heart) that have been identified as less dynamic using the training data. As already noted, we agree that the images cannot be displayed live on the scanner during the examination because of this property that there is variable time resolution across the field of view. However, once the reconstruction is complete the result is a time series of images that genuinely depict the heart beating – in fact we work out the heart rate by Fourier analysis of this time series of images. We then re-order the individual images determining cardiac phase using modulo arithmetic in time steps of the R-R interval.

After all the “real time” images have been re-mapped into a single R-R interval, we interpolate in cardiac phase across all the samples to achieve the equivalent of a conventional gated or cardiac synchronised sequence. In the final output, each cardiac phase does have data obtained over many heart beats. There is a connection here with conventional MRI sequences that use segmented acquisitions – for conventional sequences data from multiple heart beats is combined in k-space, whereas in our paper this combination occurs in the image domain. Thus we feel it appropriate to retain the terminology used and also would like to note that we have used the term ‘real-time’ in our previous publications, so this would maintain consistency.

d) The term ‘velocity volume’ and ‘velocity component volume’ and ‘velocity magnitude volume’ is repeatedly used for velocity measures, not flow measures. This seems to be an unconventional term that I can not find defined anywhere.

[R1.11d] We apologise for the confusion this terminology has clearly created and thank the referee for pointing this out – we have tried to clarify this in the manuscript and hope that the text will be clearer and easier to understand as a result. We use the term “velocity volume” because this is what the reconstruction pipeline generates. The

fundamental outputs are three separate 4D volume images (3D space + cardiac phase) which contain scalar values equal to the Cartesian components of the vector velocity at each “voxel” in space-time (cardiac phase). We refer to these as “velocity component volumes”. We can then combine them into derived 4D space-time image volumes in which each space-time “voxel” now has a 3D vector velocity associated with it (thus we have a 4D array where each array element is a 3-vector). Alternatively we may take the square-root of the sum of squares element-wise across the three velocity component volumes to produce a “velocity magnitude volume”, which is a 4D array of scalar values representing the magnitude of the velocity in each space-time “voxel”. (We placed the word voxel in quotes above as a “voxel” usually stands for a volume element – we do not know of an equivalent word for a space-time volume element).

We do appreciate that this is somewhat specialist usage, but we feel that the terms are descriptive of the data structures we work with. We have updated the manuscript to try to clarify our meaning [p 11, line 12].

Minor issues

12) The authors don't define what 4D flow MRI is until page 4 last paragraph (after abstract and 1.5 pages of intro). Yet other terms such as 4D cine imaging of the fetal heart are used. I suggest to define 4D flow MRI much earlier and in the abstract since the term '4D Flow MRI' is somewhat ambiguous. Here, a dynamic, volumetric acquisition of the 3-directional velocity vector field is captured.

[R1.12] Thank you, we have now defined 4D flow MRI in the abstract [p 2, lines 10 & 13] and defined it early in the Introduction [p 3, line 16].

13) Ref 15: source unknown

[R1.13] This has now been updated.

14) Page 3, paragraph2: '... however, these approaches have been limited to single-slice ...' – that does not seem to be true for ref 15 (see comments above for incomplete ref 15)

[R1.14] We have reworked the Introduction to more clearly explain the existing methods for PC-MRI in the fetus. [p 3, line 22]

15) Ref 16: incomplete (last author is missing)

[R1.15] This has been corrected.

16) Together with ref 16: MacDonald et al demonstrated 4D flow in the fetus of the rhesus macaque at the same time as ref 16 (PMID: 30102431)

[R1.16] We have now included this reference in the Introduction. [p3, line 25]

17) Page 4/ page 5 and/or Page 7: It should be noted that a very clever 3-directional bSSFP PC approach had been previously introduced : Santini 2009 (PMID: 19585606)

[R1.17] Thank you for sharing this useful and relevant reference. We have added it to the Introduction. [p5, line 22]

18) Page 6, Fig 1: It strikes me as odd that the figures here do not use a standard right hand coordinate system. More importantly, it seems that the labels for the axes should not be x, y , and z but a moment space (or spatial and moment space?)

[R1.18] We have updated the axes to use a right hand coordinate system [p 7]. We understand what the reviewer means regarding moment space, but for the sake of clarity to readers less familiar with velocity encoding we have chosen to leave the axes in x/y/z space. The red arrows are intended to represent the directions of the moments, denoted by the subscripts on “M”, i.e: $M_x/M_{y,x}$ etc. For bSSFP, we wanted to make it clear that the direction of encoding is oblique to the natural image coordinate axes, i.e.: across two of the normally defined spatial dimensions. We have updated the figure caption to make this clearer [p 7-8, caption].

19) Page 9: grammar in Figure 2: Background phase is estimated and subtracting ...

[R1.19] This has been corrected.

20) Page 16: Simulated Flow Phantom: what Venc was used in the simulations?

[R1.20] Thank you for spotting this omission. It was 159 cm/s, which now has been updated [p 18, line 9]. We also realise that we had mistakenly written the *in utero* Venc = 79 cm/s – apologies for this typo. We have now updated this to the correct value of 159 cm/s. The presented simulation study was designed to precisely match the *in vivo* situation. [p 20, line 23].

21) Page 18: How many slices are in one stack?

[R1.21] The number of slices per stack varies depending on the size of the fetus and the orientation of the stack, but typically ranges from 7-11 slices. This has been added to the manuscript [p 20, line 7].

22) Page 23: ‘normal spiral relationship’: find that hard to see Maybe some arrows would help?. I also believe the authors mean ‘helical’ flow which is a more common description of flow in the ascending aorta.

[R1.22] This sentence was actually referring to the intertwining nature of the aorta and pulmonary artery in Figure 7b, rather than describe a flow pattern in the aorta itself.

We agree this was unclear and have updated the relevant section of the text to emphasise the vessels and not their geometry. [p 27, line 24]

Reviewer #2 (Remarks to the Author):

Dear Madams and Sirs,

thank you for giving me the opportunity to review this interesting article entitled: “Fetal Whole-Heart Blood Flow Imaging Using 4D cine MRI”.

It is an indeed a challenge to use the magnetic resonance tomography (MRI) for cardiovascular imaging (cMRI) in the fetus in utero, especially for blood flow imaging using the MRI. The authors are aware of the main problem of cMRI, the triggering of the fetal heart beat, which is necessary for both cardiac imaging and for flow measurement within the fetus.

1) Recent studies have shown that Doppler Ultrasound Triggered flow imaging for 2D and 3D cine MRI (e.g. Kording et al. 2018, Schoennagel et al. 2019) were applicable in human fetus, which the authors have not taken in account. Furthermore, at the annual meeting of the ISMRM 2019 as well as at the RSNA 2019, the same group has shown the feasibility of 4D cine MRI, which abstracts are also available online. The authors use the so-called “slice-to-volume registration” for solution and seem to succeed in flow measurement. This method is a postprocessing method and is comparable with the metric optimized gating technique, which the authors should mention in their text.

[R2.1] We of course agree that these are important contributions. We had originally included the Kording 2018 abstract using DUS to perform 4D flow MRI, which we have now replaced with the Kording 2019 abstract which is similar but more recent [p 3, line 26]. We do reference the MOG technique in the Introduction and discuss our results in the context of MOG in the Discussion. We have now added the Schoennagel 2019 and Salehi 2019 papers which use DUS in conjunction with 2D flow MRI [p3, line 24].

2) The main weakness of this whole manuscript is the lack of gold standard. Usually the ultra-sound examinations are used in the clinical routine in obstetrics and thus should be used to compare the flow velocity within the different vessels mentioned in this manuscript.

[R2.2] We agree that there is lack of a gold standard and that this is a challenge for validating new methods of quantitative flow imaging in the fetal circulation. However, we believe this is a general problem, rather than a specific omission in our paper. Our paper is intended as a demonstration of an innovative new method – a direct comparison of our method with ultrasound would require many patients (~> 50 fetuses) for meaningful statistical comparison, so would constitute an entirely new clinical study. For the subjects in our manuscript, flow measurements within the different vessels were not acquired using ultrasound, and even if they were, drawing conclusions from such a comparison would not be possible. There are multiple reasons for this. Firstly the US data and the MRI would necessarily come from separate examinations in which the pose of the mother would be different and the state of arousal of the fetus at the time of each measurement would be completely uncontrolled. Secondly US measures of vessel flow depend on the insonation angle relative to the vessel in question and on a separate estimate of the cross sectional area of the vessel, both of which are challenging and variable in fetal US. There are numerous journal articles that attest to these problems [1-5].

It is also worth noting that quantitative flow imaging by MRI has been extensively validated and is regarded as a gold standard for adult cardiac flow imaging [6-8]. Our method relies on the same underlying physics and the same analysis concepts for converting sensitised images into velocities as the established quantitative MRI flow methods, but our approaches to data acquisition and motion correction are different in a radical way. For this reason we did a careful validation in both physical and simulated flow phantoms – and also now against a benchtop measurement using a measuring cylinder (see comment R1.7). Our method agrees with both the measuring cylinder volumetric flow rate and a conventional gold-standard MRI approach. This is presented in our manuscript.

We have now also performed a validation of the fetal blood flow measurements using internal consistency within the circulatory system (the flow in different vessels should balance as blood is incompressible), which we compare against equivalent measurements in normal late gestation fetuses conducted by Prsa et al. (2014) [9] – see Reviewers Figure R3 below. The top row shows schematics of the fetal circulation from the Prsa paper. The bottom row shows equivalent schematics created using results from our manuscript. Note, we do not measure the umbilical vein, which is measured by Prsa et al.

Reviewers Figure R3: Schematics of mean flow rates through the fetal circulation. (a,b) Reference measurements from the Prsa et al. 2014 paper in 40 normal fetuses at term (i.e.: late GA). (c,d) Equivalent measurements from our results using 4D flow MRI. The cohort was seven fetuses from 24-33 weeks GA, including some abnormal hearts with expected normal flows. Left hand column shows mean absolute flow rates averaged across the cohorts, normalised to fetal weights. Right hand

column shows flow rates represented as percentages of total inflow and total outflow. See the Prsa et al (2014) paper for full details.

The absolute values measured using our technique (Reviewers Figure R3c) are lower than those measured by Prsa et al (Reviewers Figure R3a), which we attribute to limitations of temporal and spatial resolution. We discuss these in the manuscript.

Despite the discrepancy in measured absolute values, our relative flow rate percentages are very encouraging. The MPA (Prsa = 57% vs Roberts = 53%), AAo (40% vs 44%) and SVC (29% vs 25%) all show strong agreement with the Prsa paper. The DAo has a slightly larger difference (52% vs 41%) whilst the DA is the most different (41% vs 23%). This latter finding is not surprising given the difficulties in delineating this vessel reported by our expert readers, associated with the aforementioned resolution limitations. We also see very strong agreement when we compare mean total outflow (AAo + MPA +) against mean total inflow (DAo + SVC + PBF), which should be equal: $Q_{\text{inflow}} = 309 \text{ ml/min/kg}$, $Q_{\text{outflow}} = 321 \text{ ml/min/kg}$.

We hope that these comparisons help illustrate that despite the low absolute values our measurements are internally consistent. It should also be noted that despite a cohort of 40 normal fetuses, the Prsa paper – which is considered the best reference standard for fetal PC-MRI flow measurements to date – reports very large standard deviations. This is another example of just how difficult it is to validate fetal flows and provide reliable reference values. We believe our own measurements underestimate the true flow, but our flow values still fall within the broad ranges provided in the Prsa paper.

We provide these schematics for the reviewers in the first instance. Currently, we think they are beyond the remit of the paper because it is a technical demonstration of a new method and we do not want to place undue weight on these preliminary consistency findings.

To conclude, we have updated the Discussion to explain in the manuscript why validation is so challenging. [p 37, line 6]

[1] Gill, Robert W. "Measurement of blood flow by ultrasound: Accuracy and sources of error" *Ultrasound in Med. & Biol.* Vol. 1, Issue 4, pp. 625-641, (1985)

[2] Burns, P. N. "Measuring volume flow with Doppler ultrasound—an old nut." *Ultrasound in Obstetrics and Gynecology: The Official Journal of the International Society of Ultrasound in Obstetrics and Gynecology* 2.4 (1992): 238-241.

[3] Hoyt, Kenneth, et al. "Accuracy of Volumetric Flow Rate Measurements: An In Vitro Study Using Modern Ultrasound Scanners". *J Ultrasound Med* (2009); 28:1511–1518.

[4] Jensen, Jonas, et al. "Vector velocity volume flow estimation: Sources of error and corrections applied for arteriovenous fistulas." *Ultrasonics* 70 (2016): 136-146.

[5] Vergara, Christian, et al. "Womersley number-based estimation of flow rate with Doppler ultrasound: sensitivity analysis and first clinical application." *computer methods and programs in biomedicine* 98.2 (2010): 151-160.

[6] Friedrich, Matthias G. "The Future of Cardiovascular Magnetic Resonance Imaging." *European heart journal* 38.22 (2017): 1698.

[7] Dyverfeldt, Petter, et al. "4D flow cardiovascular magnetic resonance consensus statement." *Journal of Cardiovascular Magnetic Resonance* 17.1 (2015): 72.

[8] Fogel, Mark A., ed. **Principles and practice of cardiac magnetic resonance in congenital heart disease: form, function and flow – Chapter 4: Assessment of ventricular function and blood flow.** John Wiley & Sons, 2010.

[9] Prsa, Milan, et al. "Reference ranges of blood flow in the major vessels of the normal human fetal circulation at term by phase-contrast magnetic resonance imaging." **Circulation: Cardiovascular Imaging** 7.4 (2014): 663-670.

3) The use of 4D flow measurement is very promising, but from clinical point of view this post processing, rather complicated method is not really a solution for the daily use.

We agree that currently our approach is not developed into a tool suitable for routine clinical use (hence all the code is available on Github rather than developed into a self-contained software package). The purpose of our paper is to present a cutting-edge method which does something that has hitherto been infeasible. We believe that the material presented provides compelling evidence that the approach is feasible and that this evidence can provide powerful motivation for expending the engineering effort needed to create a clinical capability. We selected this journal in the Nature publications family because it is a place where readers will seek to find out about new science and technology, rather than choosing a clinical Journal, such as the Lancet, where evidence pointing to established clinical benefit would be expected. Naturally we hope that our research will lead to substantial clinical benefit, and in fact we are already planning future studies that will aim precisely to gather evidence of clinical impact. To achieve these studies requires further engineering work, so they are some way off.

Having stated the above, it is notable that when we have shown our data to colleagues and external clinicians, we are immediately asked when they can start to use our methods. Our clinical collaborators on the manuscript ask the technical team regularly about progress towards full deployment. We estimate that there is at least two years' work needed to create a tool that others could deploy without substantial technical support. During this development phase we will be doing extensive testing in our own clinical environment to ensure the developed capability is optimised for clinical use.

4) It is also not very clear, which advances can be obtained with this rather mathematical method compared to ultrasound, which is readily available in most OB-Gyn departments.

[R2.4] Doppler ultrasound (DUS) is unquestioningly widely in clinical practice. As currently deployed it has some major strengths and some critical weaknesses. No one is expecting MRI to replace US as the first line investigation for obvious reasons of cost, convenience and availability. However, in other areas of fetal medicine there has been a progressive realisation that advanced MRI methods can be highly beneficial to patients in complex and equivocal cases [1] – our recent paper is testament to this [2]. Starting from this context, it is worth noting that what we propose in this paper delivers both where Doppler flow imaging is strong and where it is weak or has no capability. DUS can only estimate velocity and direction of flow in one vessel at a time – true vascular flow rates require a separate measurement of vessel area. Combining the flow and anatomy data is known to be highly error prone with ultrasound and as a result it is simply not used in clinical practice. We describe the reasons for this in the Introduction of our paper: DUS relies on line-of-sight velocity assessment combined with

assumptions about the shape of the targeted blood vessel and the insonation angle of the probe. These measurements are limited further by practical problems, such as fetal lie, maternal habitus and sonographer expertise. The limitations of DUS are well documented, comprehensively described by Gill in 1985 [3] and others since [4-8] – we now include these references in the Introduction [p 3, line 7]. The clinical problem is that despite continued improvement in sonography instrumentation, including Doppler, reliance on still images or cine-loops limits detection of congenital heart disease during screening [7]. Our proposed method can measure local flow velocity and direction at any chosen location in the heart and great vessels, but it also provides simultaneous measurement of flow velocity and direction of all the blood in these regions. Furthermore it provides geometrically locked anatomy and velocity maps from a single acquisition, so that volumetric flow can be robustly determined. It works regardless of fetal lie and in our paper we show how every major vessel (with the exception of the ductus arteriosus in one young fetus) could be measured.

Metric Optimized Gating (MOG) was a ground breaking technique in the area of assessing fetal flows when it was developed ten years ago, but it has major limitations in its approach to acquisition because small fetal movement compromises the accuracy of flow measurements and, at worst, major fetal motion means a new scan must be planned and acquired. Furthermore, MOG can only be used to measure flow in a 2D plane, so like DUS the clinician has to decide which specific measurements to focus on. The method we present has promise to match MOG in accuracy while achieving robustness to general movement that is not possible with a single slice approach and while providing comprehensive coverage that enables full retrospective review on all vessels and cardiac chambers

The Doppler ultrasound-gated 4D flow method by Kording et al. is a very promising development as it allows conventional gated-MRI acquisitions to be acquired in the fetus. But, the issue of fetal movement is not addressed, which is a major problem, especially in lower GA fetuses.

In conclusion, as with MRI in other emerging applications in obstetrics, the methods we present provide detailed and comprehensive data that cannot be matched by DUS or US in general. But the application of these methods is expected always to follow US examination and is only appropriately deployed in cases that are complex or where US is equivocal. As congenital heart disease is the single largest category of fetal anomaly and a distressing number of babies are still born with incompletely or incorrectly diagnosed anomalies, we believe that a robust, accurate and comprehensive fetal cardiac imaging capability can deliver substantial medical benefits. This paper is a first presentation of a first credible capability of that kind. The prior work on MOG was an astonishing achievement, but it does not achieve the critical aspects of robustness and comprehensiveness that we have now shown.

[1] Prayer, D., et al. "ISUOG Practice Guidelines: performance of fetal magnetic resonance imaging." *Ultrasound in Obstetrics & Gynecology* 49.5 (2017): 671-680.

[2] Lloyd, David FA, et al. "Three-dimensional visualisation of the fetal heart using prenatal MRI with motion-corrected slice-volume registration: a prospective, single-centre cohort study." *The Lancet* 393.10181 (2019): 1619-1627.

[3] Gill, Robert W. "Measurement of blood flow by ultrasound: Accuracy and sources of error" *Ultrasound in Med. & Biol.* Vol. 1, Issue 4, pp. 625-641, (1985)

- [4] Burns, P. N. "Measuring volume flow with Doppler ultrasound—an old nut." *Ultrasound in Obstetrics and Gynecology: The Official Journal of the International Society of Ultrasound in Obstetrics and Gynecology* 2.4 (1992): 238-241.
- [5] Hoyt, Kenneth, et al. "Accuracy of Volumetric Flow Rate Measurements: An In Vitro Study Using Modern Ultrasound Scanners". *J Ultrasound Med* (2009); 28:1511–1518.
- [6] Jensen, Jonas, et al. "Vector velocity volume flow estimation: Sources of error and corrections applied for arteriovenous fistulas." *Ultrasonics* 70 (2016): 136-146.
- [7] Vergara, Christian, et al. "Womersley number-based estimation of flow rate with Doppler ultrasound: sensitivity analysis and first clinical application." *computer methods and programs in biomedicine* 98.2 (2010): 151-160.
- [8] Scott, Ted, Judy Jones, and Hans Swan. "Screening for Congenital Heart Disease: Sonographic Features and Techniques for Prenatal Detection." *Journal of Diagnostic Medical Sonography* 32.4 (2016): 191-200.

5) It seems that there are other prospective triggering methods that are probably easier to use for the consumer.

[R2.5] We agree with the reviewer that there are other triggering methods, including the prospective MRI compatible Doppler US-triggered method. This system has the benefit of allowing MRI systems to perform standard cardiac-gated acquisitions. This has a benefit of fitting more closely with established cardiac MRI practice as performed in adults and younger subjects *ex utero*, but if the fetus moves significantly (which is common), then gated imaging protocols borrowed from adult examinations will fail. When there is fetal motion, the only alternative to repeat scanning is a fully motion-corrected method, such as described in our manuscript. Also, our method is fully compatible with prospectively-triggered acquisitions, if preferable to retrospective triggering. We have already made comparisons of flow measurements between our method and MOG. We have now expanded the Discussion to include a more thorough comparison of our method in the wider context of MOG and Doppler ultrasound-gated flow MRI [p 39, line 3].

6) To be published, the authors should include more patients.

[R2.6] As mentioned above, our paper is intended to demonstrate an innovative new technique. In common with many other researchers, we generally aim to publish a proof-of-principle paper. This initial study then provides the pilot data needed to specify a suitable examination needed to support larger scale clinical studies, which then deploy more stable, probably more refined methodology. For meaningful statistical comparison, we would need to add many, many more patients, which we do not have, and could not justify examining using exactly the examination we have demonstrated so far. We believe the distinction between technical publications that communicate advances with potential for clinical significance and clinically focused studies which communicate evidence of the clinical utility of new methods is important.

However, we thank the reviewer for raising this point as we realise that the original submission only included images from a single subject at ~24 weeks GA. We now include commentary and visualisations from another, older (~32 weeks GA) subject

which demonstrates the increase in blood volume and blood flow velocities as the heart develops and in the presence of an anomaly (right aortic arch). We include new visualisations [Figure 8], new videos [p 46, lines 16 & 20 – Supplementary Videos S4 & S5] and we update the Results [Figure 9b // p 28, line 5].

7) The authors should also cite some papers which have been dealing with the triggering method, especially, because there are some approach for fetal cardiac MRI using an MR-compatible cardio-tocograph (CTG) as an prospective, external triggering method.

[R2.7] We had originally included the Kording 2018 abstract which uses the Doppler ultrasound system for fetal cardiac-gated flow MRI. We have updated this to the 2019 abstract and now include reference to papers by Yamamura et al. and Kording et al. which also use the MR-compatible Doppler ultrasound device to gate the fetal heartbeat. [p 4, line 1].

8) The statistical analyses are not very exact and should be more precise, especially the inter-rater reliability.

[R2.8] A similar comment was also raised by reviewer #1. We agree that this was a weakness and thank the reviewer drawing our attention to it. As described above in this letter [see R1.10], we now include expert reader vessel delineation scores, mean flow repeatability analysis including intra- and inter-rater biases, and assessment of phasic behaviour in arterial flow curves. This analysis is now described in the Methods [p 22, line 1], the corresponding section of the Results has been re-written [p 31-32, line 14 onwards], Figure 9 has been updated, [p 33], Table 2 has been added [p 49] and the Discussion has been updated [p 36, line 10 // p37, line 22].

Overall, the idea behind this manuscript is highly interesting, but it needs to be worked over again, especially the clinical impact is not clear.

We thank the reviewer for this generous final comment and hope that our responses above sufficiently articulate the specific scope and structure of our paper.

REVIEWERS' COMMENTS:

Reviewer #1 (Remarks to the Author):

This manuscript describes the implementation and initial validation of a new MRI approach to conduct large volume flow sensitive MRI studies in the fetus that can overcome significant fetal motion during such lengthy scans. The framework of the novel approach is introduced and proof of principle studies are presented that include a numerical simulation, in vitro experiments with a tube and constant flow and in 7 human subjects.

The authors have been very responsive to the reviewers comments and the updated manuscript has undergone significant changes compared to its first submission. The replies to the reviewer comments were very detailed, included thorough follow ups, and substantial changes were incorporated into the manuscript, most notably including additional flow phantom validations, reader analysis, and artefact analysis as well as additions to the introduction and discussion sections.

The only rebuttal I disagree with is R1 – 11c – the use of the term ‘realtime MRI’. I understand the authors point that it is noteworthy to distinguish between acquisitions that ‘merge’ data acquired over multiple heartbeats into a single ‘cine’ display vs data showing individual heartbeats. It is also not lost on me that other authors in the MRI literature, and previous publications from the authors of this manuscript, have used the term ‘realtime’ MRI to describe the latter. However, I maintain that this is a misnomer and the historical incorrect use of the term does not make it right. I will contend that for anyone outside the MRI community, realtime imaging certainly does not describe an approach that requires a long lag, up to hours, between acquired data and displayed data, even if they properly represent data acquired in ‘real time’. Many readers of Nature Communications are not MRI experts, possibly clinicians, ultrasound or X-ray fluoroscopy researchers and others who will likely interpret the term ‘realtime MRI’ as something else than what is available here.

This revised manuscript has a lot of strengths: high level of innovation, potential high impact for research and clinical use of 4D Flow MRI in the fetus, feasibility demonstration in silico, in vitro, and in vivo, and good quality visualizations of the velocity vector fields. As such, it is a notable leap in the use of quantitative flow MRI. It lacks the demonstration of clinical impact, but that was not the purpose of this manuscript that focuses on the introduction and initial demonstration of a novel MRI methodology enabling new clinical applications.

Reviewer #2 (Remarks to the Author):

Dear Madams and Sirs,

thank you for re-submitting the manuscript with thorough answers to the stated comments and questions. the authors have managed to address every single point and the manuscript improved a lot. All the limitations are now mentioned and both major and minor issues are cleared. although the clinical relevance is not very high at this point, from the scientific point of view, this manuscript could be an important milestone in the field of fetal magnetic resonance imaging. further studies will show if this method can be improved and made easier for the customer, i.e. radiologists, for clinical routine.

Again, thank you for sharing this interesting study with us and I now feel comfortable accepting this manuscript for publication.

REVIEWERS' COMMENTS:

Reviewer #1 (Remarks to the Author):

This manuscript describes the implementation and initial validation of a new MRI approach to conduct large volume flow sensitive MRI studies in the fetus that can overcome significant fetal motion during such lengthy scans. The framework of the novel approach is introduced and proof of principle studies are presented that include a numerical simulation, in vitro experiments with a tube and constant flow and in 7 human subjects.

The authors have been very responsive to the reviewers comments and the updated manuscript has undergone significant changes compared to its first submission. The replies to the reviewer comments were very detailed, included thorough follow ups, and substantial changes were incorporated into the manuscript, most notably including additional flow phantom validations, reader analysis, and artefact analysis as well as additions to the introduction and discussion sections.

The only rebuttal I disagree with is R1 – 11c – the use of the term 'realtime MRI'. I understand the authors point that it is noteworthy to distinguish between acquisitions that 'merge' data acquired over multiple heartbeats into a single 'cine' display vs data showing individual heartbeats. It is also not lost on me that other authors in the MRI literature, and previous publications from the authors of this manuscript, have used the term 'realtime' MRI to describe the latter. However, I maintain that this is a misnomer and the historical incorrect use of the term does not make it right. I will contend that for anyone outside the MRI community, realtime imaging certainly does not describe an approach that requires a long lag, up to hours, between acquired data and displayed data, even if they properly represent data acquired in 'real time'. Many readers of Nature Communications are not MRI experts, possibly clinicians, ultrasound or X-ray fluoroscopy researchers and others who will likely interpret the term 'realtime MRI' as something else than what is available here.

[Rev. 1.1] We understand the reviewer's point on this matter – it is one we have debated internally and is discussed in the wider MRI community (see Dietz et al: doi:10.1002/mrm.27487 and Nayak: doi:10.1002/mrm.27770). For clarification, as we didn't include this in our last response, we have previously adopted the nomenclature used by Nayak – that our acquisition is "real-time", but not "real-time interactive". We agree that readers of Nature Communications may be likely to interpret "real-time" as more instantaneous, therefore we have altered the manuscript throughout to refer to the acquisition as "dynamic" rather than "real-time". We include an additional sentence in the Methods to explain that the temporal resolution is sufficient to capture real-time changes in the fetal cardiac motion without the necessity for periodic movements or gating, however image reconstruction is not instantaneous [p 27, line 23]. We also include a reference to the Nayak letter which we hope provides additional insight if the reader is interested.

This revised manuscript has a lot of strengths: high level of innovation, potential high impact for research and clinical use of 4D Flow MRI in the fetus, feasibility demonstration in silico, in vitro, and in vivo, and good quality visualizations of the velocity vector fields. As such, it is a notable leap in the use of quantitative flow MRI. It lacks the demonstration of clinical impact, but that was not the purpose of this manuscript that focuses on the introduction and initial demonstration of a novel MRI methodology enabling new clinical applications.

Reviewer #2 (Remarks to the Author):

Dear Madams and Sirs,

thank you for re-submitting the manuscript with thorough answers to the stated comments and questions. the authors have managed to address every single point and the manuscript improved a lot. All the limitations are now mentioned and both major and minor issues are cleared. although the clinical relevance is not very high at this point, from the scientific point of view, this manuscript could be an important milestone in the field of fetal magnetic resonance imaging. further studies will show if this method can be improved and made easier for the customer, i.e. radiologists, for clinical routine.

Again, thank you for sharing this interesting study with us and I now feel comfortable accepting this manuscript for publication.

We are pleased that both reviewers found the work interesting and we are very grateful to both reviewers for taking the time to provide feedback on our paper, especially given the difficulties caused by the ongoing pandemic.